# Grounding Video Models to Actions through Goal Conditioned Exploration

**Yunhao Luo[†‡], Yilun Du[§]**
Georgia Tech[†], Brown[‡], Harvard[§]
https://video-to-action.github.io/

## Abstract

Large video models, pretrained on massive amounts of Internet video, provide a rich source of physical knowledge about the dynamics and motions of objects and tasks. However, video models are not grounded in the embodiment of an agent, and do not describe how to actuate the world to reach the visual states depicted in a video. To tackle this problem, current methods use a separate vision-based inverse dynamic model trained on embodiment-specific data to map image states to actions. Gathering data to train such a model is often expensive and challenging, and this model is limited to visual settings similar to the ones in which data are available. In this paper, we investigate how to directly ground video models to continuous actions through self-exploration in the embodied environment – using generated video states as visual goals for exploration. We propose a framework that uses trajectory level action generation in combination with video guidance to enable an agent to solve complex tasks without any external supervision, e.g., rewards, action labels, or segmentation masks. We validate the proposed approach on 8 tasks in Libero, 6 tasks in MetaWorld, 4 tasks in Calvin, and 12 tasks in iThor Visual Navigation. We show how our approach is on par with or even surpasses multiple behavior cloning baselines trained on expert demonstrations while without requiring any action annotations.

## 1 Introduction

Large video models (Brooks et al., 2024; Girdhar et al., 2023; Ho et al., 2022) trained on a massive amount of Internet video data capture rich information about the visual dynamics and semantics of the world for physical decision-making. Such models are able to provide information on how to accomplish tasks, allowing them to parameterize policies for solving many tasks (Du et al., 2024). They are further able to serve as visual simulators of the world, allowing simulation of the visual state after a sequence of actions (Brooks et al., 2024; Yang et al., 2024c), and enabling visual planning to solve long-horizon tasks (Du et al., 2023).

However, directly applying video models zero-shot for physical decision-making is challenging due to embodiment grounding. While generated videos provide a rich set of visual goals for solving new tasks, they do not explicitly provide actionable information on how to reach each visual goal. To ground video models to actions, existing work has relied on training an inverse dynamics model or goal-conditioned policy on a set of demonstrations from the environment (Black et al., 2023; Du et al., 2024; 2023). Such an approach first requires demonstrations to be gathered in the target environment and embodiment of interest, which demands human labor or specific engineering (e.g. teleoperation or scripted policy). In addition, the learned policies may not generalize well to areas in an environment that are out-of-distribution of the training data.

Recently, Ko et al. (2023) proposes an approach to directly regress actions from video models, without requiring any action annotations. In Ko et al. (2023), optical flow between synthesized video frames is computed and used, in combination with a depth map of the first image, to compute a rigid transform of objects in the environment. Robot actions are then inferred by solving for actions that can apply the specified rigid transform on an object. While such an approach is effective on a set of evaluated environments, it is limited in action resolution as the inferred object transforms can be imprecise due to both inaccurate optical flow and depth, leading to a relatively low success rate in

evaluated environments (Ko et al., 2023). In addition, it is difficult to apply this approach to many robotic manipulation settings such as deformable object manipulation, where there are no explicit object transforms to compute.

We propose an alternative manner to directly ground a video model to actions *without using annotated demonstrations*. In our approach, we learn a goal-conditioned policy, which predicts the actions to reach each synthesized frame in a video. We learn the policy in an online manner, where given a specified task, we use each intermediate synthesized frame as a visual goal for a goal-conditioned policy from which we obtain a sequence of actions to execute in an environment. We then use the image observations obtained from execution in the environment as ground-truth data to train our goal-conditioned policy. We illustrate our approach in Figure 1.

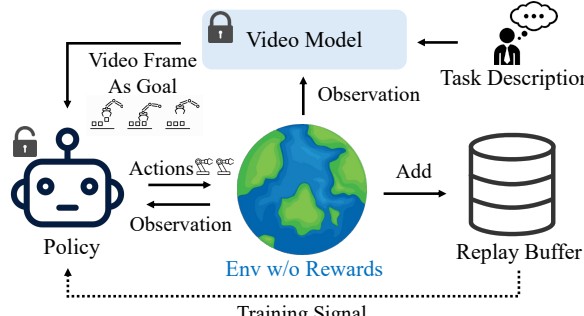

Figure 1: **Grounding Video Models to Actions.** Our approach learns to ground a large pretrained video model into continuous actions through goal-directed exploration in the environment. Given a synthesized video, a goal-conditioned policy attempts to reach each visual goal in the video, with data in the resulting real-world execution saved in a replay buffer to train the goal-conditioned policy.

In practice, directly using synthesized images as goals for exploration often leads to insufficient exploration. Agents often get stuck in particular parts of an environment, preventing the construction of a robust goal-conditioned policy. To further improve exploration, we propose to generate chunks of actions to execute in an environment given a single visual state. By synthesizing and executing a chunk of actions we can explore the environment in a more directed manner, enabling us to achieve a more diverse set of states. We further intermix goal-conditioned exploration with random exploration to further improve exploration.

Overall, our approach has three contributions: **(1)** We propose goal-conditioned exploration as an approach to ground video models to continuous actions. **(2)** We propose a set of methods for effective exploration in the environment, using a combination of chunked action prediction for temporally coherent exploration along with periodic randomized actions for robust state coverage. **(3)** We illustrate the efficacy of our approach on a set of simulated manipulation and navigation environments.

## 2 RELATED WORK

**Video Model in Decision making** A large body of recent work has explored how video models can be used in decision making (Yang et al., 2024d; McCarthy et al., 2024). Prior work has explored how video models can act as reward functions (Escontrela et al., 2024; Huang et al., 2023; Chen et al., 2021; Luo et al., 2024), representation learning (Seo et al., 2022; Wu et al., 2023; 2024; Yang et al., 2024b), policies (Ajay et al., 2024; Du et al., 2024; Liang et al., 2024), dynamics models (Yang et al., 2024c; Du et al., 2024; Zhou et al., 2024b; Brooks et al., 2024; Rybkin et al., 2018; Mendonca et al., 2023), and environments (Bruce et al., 2024). Our work explores that given video generations, how we can learn a policy and infer the actions to execute in an environment *without any action labels*, while existing work (Baker et al., 2022; Bharadhwaj et al., 2024a; Black et al., 2023; Wen et al., 2023; Zhou et al., 2024a; Wang et al., 2024) requires domain specific action data. Most similar to our work is AVDC (Ko et al., 2023), which uses rigid object transformations computed by optical flow between video frames as an approach to extract actions from generated videos. We propose an alternative unsupervised approach to ground video models to continuous action by leveraging video-guided goal-conditioned exploration to learn a goal-conditioned policy.

**Learning from Demonstration without Actions.** A flurry of work has studied the problem of robot learning from demonstrations without actions. One line of work studies the problem of extrapolating the control actions assuming that the expert state trajectories are provided (Torabi et al., 2018; Radosavovic et al., 2021; Li et al., 2024), though collecting such ground-truth state-level data is expensive and hard to scale. Some recent work explores learning from image/video robotic data without actions (Seo et al., 2022; Wu et al., 2024; Mendonca et al., 2023; Ma et al., 2022; Wang et al., 2023; Ma et al., 2023; Schmeckpeper et al., 2021), either by constructing a world model,

reward model, or representation model. However, these methods usually need additional finetuning on data or rewards from specific downstream environments. By contrast, our method uses action-free demonstration videos to train a video generative model and leverages the generated frames as goals to learn a goal-conditioned policy to complete a task. As a result, our policy learning requires neither action labels nor environment awards which can be challenging to obtain.

**Robotic Skill Exploration.** Typical robot skill exploration is formulated as an RL problem (Haarnoja et al., 2018; Hafner et al., 2019) and assumes some form of environment rewards, but the design of reward functions is highly task dependent and demands human labor. To this end, some recent work explores robotic skill exploration without any rewards. One typical class of methods is developed upon computing intrinsic exploration rewards to promote rare state visit. These methods can be based on prediction error maximization (Pathak et al., 2017; Henaff, 2019; Shyam et al., 2019), disagreement of an ensemble of world models (Hu et al., 2022; Sekar et al., 2020; Sancaktar et al., 2022), entropy maximization (Pong et al., 2020; Pitis et al., 2020; Jain et al., 2023; Eysenbach et al., 2019), counting (Bellemare et al., 2016), or relabeling (Ghosh et al., 2021). However, discrepancy, counting, or relabeling based methods are limited to simple environments, short-horizon tasks, or low-dimensional state-space. We propose to use video models for direct exploration, as large pretrained video models are a rich source of task-specific information. With efficient guidance from the video model, we show that our method is able to collect high-quality data from the environment and accomplish challenging long-horizon tasks conditioned on language instruction at test time.

## 3 METHOD

In this section, we describe our method to ground video models to actions by unsupervised exploration in the environments. First, in Section 3.1, we describe the pipeline of policy execution conditioned on video frame and hindsight relabeling via environment rollouts. Next, in Section 3.2, we introduce a periodic random action bootstrapping technique to secure the quality of video-guided exploration. Finally, in Section 3.3, we propose a chunk-level action prediction technique to further enhance the coverage, stability, and accuracy of the goal-conditioned policy. The pseudocode of our method is provided in Algorithm 1.

### 3.1 LEARNING GOAL-CONDITIONED POLICIES THROUGH HINDSIGHT RELABELING

A key challenge in unsupervised skill exploration is that the possible underlying environment states are enormous, making it difficult for an agent to discover and be trained on every valid state, especially in the high-dimensional visual domain. Many relevant states for downstream task completion, such as stacking blocks on top of each other or opening a cabinet, require a very precise sequence of actions to obtain, which is unlikely to happen from random exploration.

Video models have emerged as a powerful source of prior knowledge about the world, providing rich information about how to complete various tasks from large-scale internet data. We leverage the knowledge contained in these models to help guide our exploration in an environment to solve new tasks. To this end, we propose a novel method that uses a pre-trained video model $f_\theta(x_{\text{start}}, c)$ to guide the exploration and shrink the search space only to task-relevant states ($x_{\text{start}}$ denote the initial observation and $c$ denote the corresponding task description). This concurrently benefits both sides: the goal-conditioned policy obtains task-relevant goals so as can perform efficient exploration centered around the task-relevant state space; on the other hand, the underlying information from the video model is extracted to end effectors and enables effective decision-making and control in embodied agents. Specifically, during exploration, we first leverage the video model to generate a sequence of images based on the given image observation $x_{\text{start}}$ and language task description $c$,

$$\texttt{pred\_v} = f_\theta(x_{\text{start}}, c), \quad \text{where } x_{\text{start}} \sim \mathcal{O} \text{ and } c \sim \mathcal{T} \tag{1}$$

where $\mathcal{O}$ is the observation space and $\mathcal{T}$ is the task space. We then execute the actions predicted by the goal-conditioned policy $a^{\text{pred}} = \pi(a|x_{\text{start}}, x_{\text{goal}})$ in the environment, where the goal $x_{\text{goal}}$ is set to each predicted video frame $\texttt{pred\_v}_i$ sequentially. Hence, we can obtain an episode of image action pairs $(x_{1:t}^{\text{env}}, a_{1:t}^{\text{pred}})$ of length $t$ from the environment and these rollout data will be added to a replay buffer $R$ for policy training.

At the beginning of training, these data might not necessarily reach the exact goals given by the video model, but are still effective to indicate the task-specific region, since the rollout results of $a_{1:t}^{\text{pred}}$ are relabeled and can reflect the ground-truth environment dynamics. Empirically, we observe the

---

**Algorithm 1** Grounding Video Model to Actions

---

1: **Require:** a frozen video diffusion model $f_\theta(x_{\text{start}}, c)$, a goal conditioned policy to train $\pi(a|x_{\text{start}}, x_{\text{goal}})$, a replay buffer $R$
2: **Hyperparameters:** horizon of policy $\pi$ $h$, training iteration $N$, number of initial / additional random action episodes $n_r$ / $n_r'$, video-guided / random action exploration frequency $q_v$ / $q_r$
3: Sample $n_r$ episodes of random actions and add to replay buffer $R$
4: **for** $i = 1 \rightarrow N$ **do**
5:     **if** $i \mod q_v == 0$ **then**                 *# conduct video-guided exploration with frequency $q_v$*
6:         Sample a task $c'$, obtain observation $x_0$, and generate video $\texttt{pred\_v} = f_\theta(x_0, c')$
7:         Execute policy $\pi(a|\cdot, \cdot)$ in the environment where the goals are frames from $\texttt{pred\_v}$
8:         Add the resulting image-action pairs from the video rollout to replay buffer $R$
9:     **end if**
10:     **if** $i \mod q_r == 0$ **then**         *# conduct periodic random action bootstrapping with frequency $q_r$*
11:         Sample additional $n_r'$ episodes of random actions and add to replay buffer $R$
12:     **end if**
13:     *# train the policy with the data sampled from the replay buffer*
14:     $(x_{i:i+h}, a_{i:i+h})$ = sample a consecutive sequence of image-action pairs from replay buffer $R$
15:     $a^{\text{pred}} = \pi(a|x_i, x_{i+h})$
16:     $\mathcal{L} = \text{MSE}(a^{\text{pred}}, a_{i:i+h})$
17: **end for**
18: **return** goal-conditioned policy $\pi$

---

proposed video-guided exploration scheme is very effective, and we visualize the curve of number of success versus total number of video-guided rollouts in Figure 8. Our overall training objective is

$$\max_\phi \; \mathbb{E}_{(x_{i:i+h}, a_{i:i+h}) \sim R} \big[ \log \pi_\phi(a_{i:i+h}|x_i, x_{i+h}) \big] \tag{2}$$

where $i$ is a randomly sampled temporal index inside a rollout episode, $h$ denotes the horizon of the policy, and $\phi$ represents the parameters of the goal-conditioned policy. Note that provided a video model $f_\theta$ trained on internet scale data, the video predictions can be easily generalized to a broad observation space $\mathcal{O}$ and task space $\mathcal{T}$, enabling the proposed guidance scheme directly applied to various unseen scenarios.

## 3.2 PERIODIC RANDOM ACTION BOOTSTRAPPING

When a goal-conditioned policy is initialized from scratch, it is unable to effectively process the frames provided by a video model and use them to guide exploration, as it is unable to process input frames. Empirically, we found that a randomly initialized policy would often instead preferentially output particular actions (i.e. only move up) independent of goals given by the video model.

This significantly compromises the exploration because the policy will not explore the task-related states as we expect. More importantly, though we can obtain ground-truth environment dynamics via hindsight relabeling, the actions in the replay buffer $R$ are output from the scratch policy itself. This might result in an undesirable loop where only the previous outputs are used as the ground-truth and these actions are irrelevant to completing the task.

To this end, inspired by random action sampling in RL setting (Sutton, 2018), we propose a novel periodic random action bootstrapping method for grounding video model to actions. Specifically, we first conduct random action exploration and append the resulting data to the replay buffer $R$ before the training starts, and periodically conduct extra random exploration during training. The process can be denoted by

$$R \; \leftarrow \; a_{1:t^r}^{i_r} \sim \texttt{Uniform}\big[a_{\text{low}}, a_{\text{high}}\big] \quad \text{for } i^r \text{ in } [1, 2, ..., n_r] \tag{3}$$

where $n_r$ denotes the number of random action episodes, $t^r$ denotes the length of a random action episode, $a_{\text{low}}$ and $a_{\text{high}}$ are the action limits. This proposed bootstrapping method can enhance the exploration in two ways: the initial random actions serve as the basic world dynamics information which enables the policy to reach the vicinity of goal states specified by video frames, ensuring effective video-guided exploration; the periodic extra random exploration can further expand the agent's discovered state space and stabilize policy training. Please see Section 4.3 for ablation studies and Appendix D.3 for implementation details.

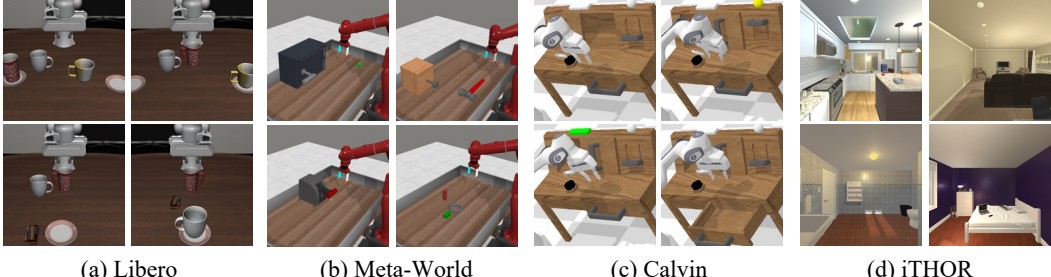

|        (a) Libero        |     (b) Meta-World     |       (c) Calvin       |       (d) iTHOR        |

Figure 2: **Environment Demonstrations.** We evaluate our method on three robotic manipulation environments: Libero, Meta-World, Calvin, and one visual navigation environment: iThor, with a total of 30 tasks. Images in (a) and (c) denote the goal object states of a subset of tasks. Images in (b) are randomly sampled start observations of a subset of tasks. In (d), we show the layout for each scene from the agent's view.

## 3.3 Chunk Level Action Prediction and Exploration

In current robot exploration frameworks, the policy usually predicts a single action as output and explores the environment with the predicted action. However, single-action prediction can hinder diverse exploration, as the agent can easily stuck in a single state, outputting the same repeated action. In addition, trajectories generated in such manner are often incoherent, as repeatedly sampled actions may reverse and undo actions generated previously in the trajectory. Finally, exploration with single action prediction requires one model forward pass for each action, resulting in a larger computational burden and higher latency.

To tackle these issues, we propose to predict a sequence of actions with horizon $h$ and explore the environments using the predicted chunks of actions. On one hand, by modeling behavior over a longer horizon, chunk-level action prediction can encourage more coherent action sequences and mitigate myopic actions and compounding errors, especially when the temporal distance to the goal is large. On the other hand, exploration with action chunks can in turn collect more coherent data from the environment, further facilitating the policy to capture the underlying environment dynamics. In addition, we utilize chunk-level actions during random exploration. Specifically, we sample an action mean $a^m$ from a uniform distribution, and based on $a^m$, we sample a chunk of actions $a^c$ of length $l_c$ from Gaussian distribution, where the $i$-th action is represented as $a_i^c \sim \mathcal{N}(a^m, \sigma)$. This ensures consistent exploration and avoids the zero-mean random action issue that confines the agent to a small vicinity near the start state.

Recent works have also employed action chunking (Bharadhwaj et al., 2024b; Zhao et al., 2023) in behavior cloning. However, our application is different and we propose to use action chunking to enable more effective unsupervised exploration. We illustrate how, in combination with video-guided exploration, this action chunking allows for more coherent exploration, as well as enabling models to make consistent plans to achieve video goals. In Section 4.3, we provide a study comparing chunk-level prediction versus single action prediction in the Libero environment and empirically show that the chunk-level design can substantially improve the resulting goal-reaching policy.

## 4 Experiments

We present our experiment results across four simulated environments shown in Figure 2. In Section 4.1, we describe our results on three robotic manipulation environments: 8 tasks on Libero (Liu et al., 2024), 6 tasks on MetaWorld (Yu et al., 2020), and 4 tasks on Calvin (Mees et al., 2022). Following this, in Section 4.2, we show evaluation results on 4 different scenes and 12 targets on iTHOR (Kolve et al., 2017) visual navigation environment. Finally, in Section 4.3, we present ablation studies on the proposed chunk level action prediction and random action bootstrapping methods. We provide more experiment results in Appendix A and B. We use the same demonstration data to train the baseline methods and our video model, *except that* training the video model only requires the image sequences of demonstrations, while most baseline methods need the corresponding action annotations (highlighted by an asterisk). In evaluation, for manipulation tasks, the initial robot state and object positions are randomized; for visual navigation tasks, we randomize the robot start position.

**Implementation.** For the video model, the inputs are the observation image and the task description. We follow the lightweight video model architecture introduced in (Ko et al., 2023) and train one video model from scratch for each environment. We deem that finetuning large text-conditioned

| | put-red-mug-left | put-red-mug-right | put-white-mug-left | put-Y/W-mug-right | Overall |
|---|---|---|---|---|---|
| BC* | 8.8±5.3 | 15.2±7.8 | 32.0±12.9 | 21.6±12.3 | 19.4±9.6 |
| GCBC* | 2.4±2.0 | 0.8±1.6 | 16.0±7.2 | 7.2±5.3 | 6.6±4.0 |
| DP BC* | 33.6±3.2 | 33.6±8.2 | 59.2±7.8 | 57.6±5.4 | 46.0±6.2 |
| DP GCBC* | 24.8±4.7 | 22.4±7.4 | 16.0±8.8 | 3.2±3.0 | 16.6±6.0 |
| SuSIE* | 18.4±2.0 | 32.0±8.4 | 43.2±4.7 | 25.6±11.5 | 29.8±6.6 |
| AVDC | 0.0±0.0 | 0.0±0.0 | 0.0±0.0 | 0.0±0.0 | 0.0±0.0 |
| Ours w/ SuSIE | 23.2±3.0 | **60.0**±6.7 | **68.8**±4.7 | **67.2**±8.9 | **54.8**±5.8 |
| Ours | **38.4**±15.3 | 40.8±7.8 | 51.2±3.9 | 38.4±8.6 | 42.2±8.9 |

| | put-choc-left | put-choc-right | put-red-mug-plate | put-white-mug-plate | Overall |
|---|---|---|---|---|---|
| BC* | 19.2±9.3 | 12.8±9.3 | 7.2±5.3 | 20.0±11.3 | 14.8±8.8 |
| GCBC* | 4.8±1.6 | 4.0±4.4 | 2.4±3.2 | 7.2±6.4 | 4.6±3.9 |
| DP BC* | 42.4±5.4 | 50.4±5.4 | 32.8±9.3 | 71.2±5.3 | 49.2±6.4 |
| DP GCBC* | 45.6±6.0 | 32.0±8.8 | 7.2±4.7 | 5.6±4.1 | 22.6±5.9 |
| SuSIE* | 17.6±9.3 | 32.8±9.9 | 16.0±2.5 | 10.4±4.1 | 19.2±6.5 |
| AVDC | 1.3±1.9 | 0.0±0.0 | 0.0±0.0 | 0.0±0.0 | 0.3±0.5 |
| Ours w/ SuSIE | 44.0±7.6 | 54.4±5.4 | 66.4±12.0 | **36.0**±7.6 | 50.2±8.2 |
| Ours | **70.4**±12.8 | **79.2**±3.9 | **72.8**±6.4 | 25.6±11.5 | **62.0**±8.7 |

Table 1: **Quantitative results on 8 tasks of two different scenes in Libero.** Note that methods marked with an asterisk '*' require ground-truth action demonstrations to train, while other methods do not. *Ours w/ SuSIE* uses an image-editing model to generate subgoals guidance while *Ours* uses a video model.

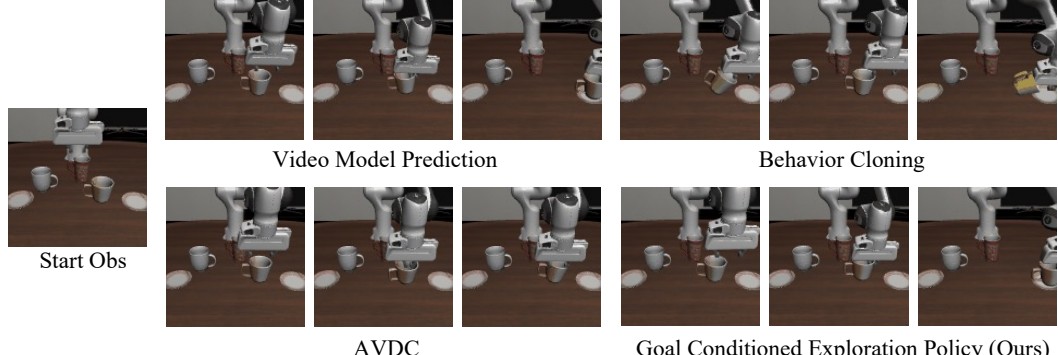

Figure 3: **Qualitative Results of task 'put the yellow and white mug to the right plate' in Libero environment.** The start states of the robot and objects are randomized in test time. Only a subset of the predicted video frames are shown due to the space limit. Our goal-conditioned policy shown in the bottom right is able to follow the video prediction and finish the task. BC cannot accurately locate the target while AVDC can move to the mug but without the skill of grasping concave objects.

video models (Yang et al., 2024e; Chen et al., 2024; Saharia et al., 2022) or designing task-specific video model might be an interesting future research direction. For the goal-conditioned policy, we implement it with a CNN-based Diffusion Policy (Chi et al., 2023), which takes as input the observation image and the goal image and outputs a chunk of actions. For detailed implementation of our method and each baseline, please refer to Appendix D.

## 4.1 MANIPULATION

In this section, we aim to evaluate the goal-conditioned policy learned by the proposed unsupervised exploration in tabletop manipulation environments with continuous action space. To better understand the capability of the method, we investigate multi-task policy learning in Libero and Calvin, and single-task policy learning in Meta-world. Note that methods highlighted by an *asterisk* require ground-truth action labels to train, whereas our method only requires image demonstration sequences to train the video model and self-supervised training for the policy model.

**Libero** (Liu et al., 2024) is a tabletop simulation of a Franka Robot, which features several dexterous manipulation tasks. For each task in Libero, the agent is required to achieve the final state described in a corresponding sentence, which identifies the target object and task completion state. The action space consists of the delta position and orientation of the end effector and the applied force on the

|  | door-open | door-close | handle-press | hammer | assembly | faucet-open | Overall |
|---|---|---|---|---|---|---|---|
| BC* | 64.0±4.4 | 76.0±0.0 | 49.6±3.2 | 4.8±3.9 | 8.0±2.5 | 88.8±3.0 | 48.5±2.8 |
| GCBC* | 64.8±6.9 | 93.6±2.0 | 50.4±6.5 | 4.8±1.6 | 8.0±2.5 | **95.2**±1.6 | 52.8±3.5 |
| DP BC* | 73.6±4.8 | 94.4±2.0 | 52.0±8.4 | 7.2±3.0 | 0.8±1.6 | 82.4±2.0 | 51.7±3.6 |
| DP GCBC* | 68.0±0.0 | 96.0±4.0 | 36.0±0.0 | 14.0±2.0 | 5.6±2.0 | 80.0±0.0 | 49.9±1.3 |
| AVDC | 52.0±0.0 | **97.3**±1.9 | 76.0±5.7 | 4.0±3.3 | 8.0±0.0 | 66.7±5.0 | 50.7±2.6 |
| Ours | **76.0**±4.9 | 85.6±2.0 | **94.4**±2.0 | **43.2**±6.4 | **16.8**±3.0 | 87.2±3.0 | **67.2**±3.6 |

Table 2: **Quantitative Results of 6 tasks in Meta-World.** Methods marked with '*' requires ground-truth action demonstrations to train. AVDC uses the segmentation mask of the target object to compute actions. Our method learns the policy by video-guided self-exploration in the environment without any external supervision.

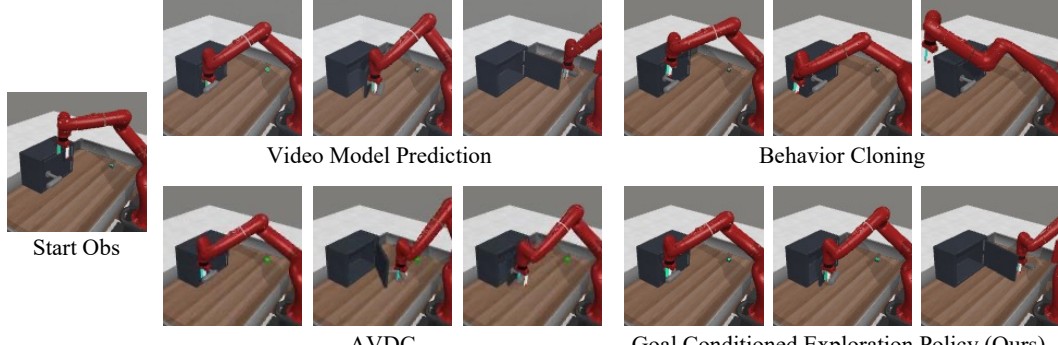

Figure 4: **Qualitative Results of task Door-Open in Meta-World environment.** The position of the box and robot are randomized in test-time. Only a subset of the predicted video frames are shown due to space limit. Our goal-conditioned policy can follow the subgoals given by the video frames and successfully finish the task. BC misses the handle probably due to the out of training distribution box position and starts to predict random actions. AVDC can move to the handle thanks to the exact given handle location. However, it begins to close the door halfway, probably because of the incorrect flow prediction due to error accumulation or occlusion.

gripper, resulting in a total dimension of 7. To improve the efficiency of exploration, we further add an exploration primitive for grasping, which we discuss in detail in Appendix D.2.

We include 8 tasks from two scenes in Libero as the testbed. In this environment, we aim to evaluate the multi-task learning capability of our method, hence we only train one policy model for all 8 tasks. The video model is trained on the visual image sequences of the demonstrations provided in Libero, where we use 20 episodes per task, thereby 160 demonstrations in total. We train BC, GCBC, SuSIE (Black et al., 2023) on the same visual image sequences but with expert actions corresponding to each image. Following the setting in AVDC, we provide it with the segmentation masks of the target objects, while our method does not require such privileged information.

We present the quantitative results in Table 1 and qualitative results in Figure 3 and 6. Across 8 tasks, our method is able to outperform baselines by a margin, though without any access to expert action data. We observe that BC-based methods tend to memorize and overfit to the training data, where they fail to locate the target objects, while our method is more robust in test-time. In addition, though knowing the exact positions of target objects in test-time, AVDC is unable to achieve any meaningful results, probably because that AVDC uses hard-coded action primitives and planning procedural, making it difficult to generalize to more complex manipulation tasks, for example, grasping concave objects such as mugs. Note that our method can also be integrated with various forms of generative models, as shown by *Ours w/ SuSIE* where we use an image-editing model to predict the subgoals for exploration. Please refer to Appendix A for more results and Appendix C for failure analysis.

**Meta-World** (Yu et al., 2020) is a simulated robotic benchmark of a Sawyer robot arm with a set of tabletop manipulation tasks that involve various object interactions and different tool use. The action space consists of the delta position of the end effector and the applied force on the gripper, resulting in a total dimension of 4.

We consider 6 tasks: door-open, door-close, handle-press, hammer, assembly, faucet-open. We directly utilize the video model checkpoint provided in AVDC. To further validate the proposed exploration approach, we conduct single-task exploration in this environment, where we train one policy model for each task. Each learning-based baseline model is trained on the same demonstrations

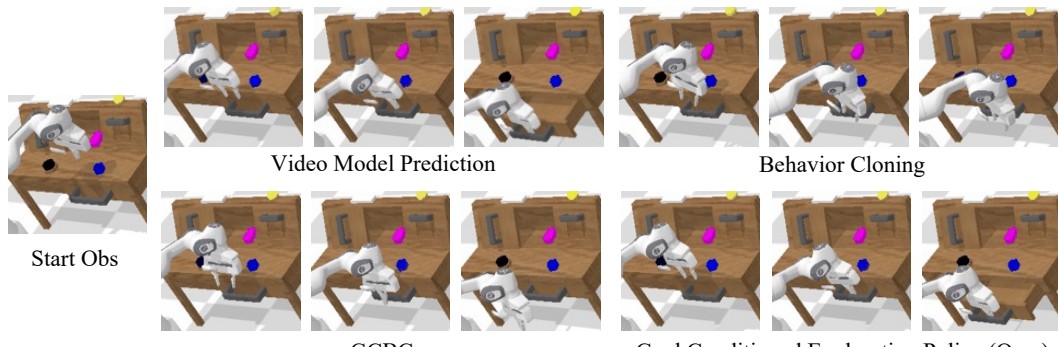

Video Model Prediction          Behavior Cloning

Start Obs

GCBC          Goal Conditioned Exploration Policy (Ours)

Figure 5: **Qualitative Results of task Open Drawer in Calvin environment.** Our goal conditioned policy successfully completes the task by following the subgoals in the video, while both BC and GCBC cannot put the end effector to the handle and thus fail to open the drawer.

| | lightbulb | led | slider left | open drawer | Overall |
|---|---|---|---|---|---|
| BC* | 47.2±21.1 | 48.8±9.9 | 67.2±14.2 | 36.0±18.8 | 49.8±16.0 |
| GCBC* | 1.6±2.0 | 38.4±13.0 | 32.0±6.2 | 22.4±7.8 | 23.6±7.3 |
| DP BC* | 70.4±2.0 | 79.2±3.9 | 68.8±4.7 | 56.8±1.6 | 68.8±3.0 |
| DP GCBC* | 35.2±3.0 | 44.0±5.7 | 40.0±3.6 | 17.6±7.4 | 34.2±4.9 |
| Ours | **100.0±0.0** | **86.4±5.4** | **83.2±3.0** | **68.0±3.6** | **84.4±3.0** |

Table 3: **Quantitative Results of 4 Calvin tasks**. Each task requires the agent to manipulate objects located in different regions, especially in open drawer where the policy needs to cover the bottom right boundary of the environment.

Figure 6: **Qualitative Comparison of our method and GCBC on Libero.**

used to train the video model, namely, 15 episodes per task. We present the quantitative results in Table 2 and qualitative results in Figure 4. We observe that our policies can successfully learn various manipulation skills and outperform all the counterparts in the average success rate despite the absence of action labels for training. In addition, we also compare with training a single-task RL with a zero-shot reward function in Appendix A.7.

**Calvin** (Mees et al., 2022) is a robotic simulation environment with multiple language-conditioned tasks. This environment contains a 7-DOF Panda robot arm and various assets including a desk with a sliding door, a drawer, an LED, and a lightbulb. The agent is required to complete tasks in the environment given by a corresponding language description. The action space we use is same as Libero.

We consider 4 tasks: turn on lightbulb, turn on led, move slider left, and open drawer. These tasks involve different manipulation skills and different operation areas in the workspace, allowing us to validate skill learning and spatial coverage capability of our goal-conditioned policy. For example, to move the slider left, the agent has to drag the handle from the central area to the upper left corner. We present the quantitative results in Table 3 and qualitative results in Figure 5. We do not report AVDC in the above table, as we found it performed very poorly in the above environments. Even without any action data or rewards, our policy is able to cover the majority of the space and outperform all the baselines, especially for the challenging open drawer task, where the agent must move downwards below the table and pull toward the bottom right boundary of the environment.

## 4.2 VISUAL NAVIGATION

In addition to tabletop-level robot arm manipulation tasks, we evaluate the proposed method in a room-level visual navigation setting with discrete actions.

**iTHOR** (Kolve et al., 2017) is a room-level vision-based simulated environment, where agents can navigate in the scenes and interact with objects. We adopt the iTHOR visual object navigation benchmark, where agents are required to navigate to the specific type of objects given by a natural language input. The action space consists of four actions: Move Ahead, Turn Left, Turn Right, and Done. We incorporate 4 different scene types: Kitchen, Living Room, Bedroom, and Bathroom, with 3 targets in each scene. Following (Ko et al., 2023), the video model is trained on 20 demonstration image sequences per task. BC and GCBC are also trained on the same demonstrations while with access to corresponding actions. Since the navigation actions are discrete and episode lengths

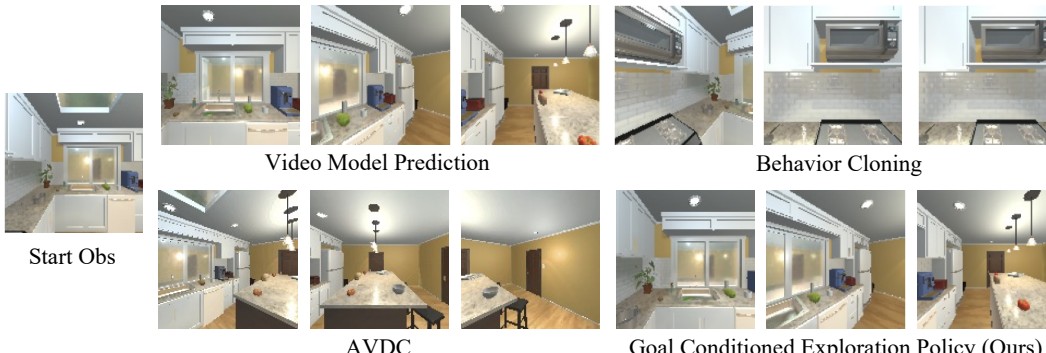

Figure 7: **Qualitative Results of Navigation to Bread in iTHOR environment.** Only a subset of the predicted video frames are shown due to the space limit. In navigation tasks with a moving camera, the video model can still generate realistic frames that respect the actual scene layout. Our goal-conditioned policy is able to follow the video frames and reach the bread shown in the middle of the frame. However, BC turns to the other way, and AVDC cannot correctly infer the required actions and miss the target.

|        | Kitchen | | | | Living Room | | | |
|--------|---------|--------|--------|---------|-------------|--------|------------|----------|
| Method | Toaster | Spatula | Bread | Overall | Painting | Laptop | Television | Overall |
| BC*    | 48.0±2.4 | **56.0**±3.7 | **36.0**±6.6 | 46.7±4.3 | **36.0**±12.4 | 45.0±11.0 | 49.0±4.9 | 43.3±9.4 |
| GCBC*  | **52.0**±5.1 | 49.0±6.6 | 39.0±5.8 | 46.7±5.9 | 27.0±8.1 | **49.0**±5.8 | 58.0±8.7 | **44.7**±7.6 |
| AVDC   | 10.0±4.1 | 13.3±4.7 | 13.3±2.4 | 12.2±3.7 | 8.3±2.4 | 13.3±8.5 | 20.0±4.1 | 13.9±5.0 |
| Ours   | 45.0±3.2 | **56.0**±4.9 | **44.0**±5.8 | **48.3**±4.6 | 29.0±4.9 | 42.0±6.0 | **57.0**±5.1 | **42.7**±5.3 |

|        | Bedroom | | | | Bathroom | | | |
|--------|---------|----------|--------|---------|----------|------------|---------|---------|
| Method | Blinds | DeskLamp | Pillow | Overall | Mirror | ToiletPaper | SoapBar | Overall |
| BC*    | **67.0**±2.4 | **40.0**±7.1 | **81.0**±5.8 | **62.7**±5.1 | **48.0**±6.8 | 51.0±13.2 | 45.0±4.5 | 48.0±8.1 |
| GCBC*  | 52.0±8.1 | 22.0±2.4 | 72.0±6.0 | 48.7±5.5 | 43.0±16.0 | 64.0±9.2 | **55.0**±11.8 | **54.0**±12.3 |
| AVDC   | 30.0±4.1 | 13.3±4.7 | 36.7±2.4 | 26.7±3.7 | 10.0±4.1 | 1.7±2.4 | 6.7±2.4 | 6.1±2.9 |
| Ours   | 57.0±5.1 | 24.0±3.7 | 72.0±7.5 | 51.0±5.4 | 36.0±9.2 | **75.0**±4.5 | 47.0±7.5 | 52.7±7.0 |

Table 4: **Quantitative Results on iThor Navigation.** We report the success rates across 4 different scenes and 12 targets. Though without access to action labels, our method is on par with the BC and GCBC trained on expert demonstrations and outperforms AVDC which also does not require expert action labels.

(typically < 20) are much shorter than robot arm manipulation tasks, we find that predicting a single action suffices. Our method uses the same ResNet+MLP model architecture as BC-based baselines.

We present the quantitative results in Table 4. Our method outperforms the no expert data baseline AVDC by 36% on average and surpasses all baselines in the Kitchen scene while performing on par with BC-based baselines in other scenes. Qualitative results are shown in Figure 7 and 16. Our goal-conditioned policy is able to reliably synthesize discrete actions that follow the generated video plan. In contrast, the actions inferred by AVDC are incorrect: the agent keeps rotating right and misses the target, whereas the underlying actions in the video should be moving ahead and then rotating right, probably because that the drastic change in observation poses challenges for optical flow prediction. BC predicts a wrong action at the first step, where the agent directly rotates to the opposite side and navigates to the target Spatula, likely due to its limited generalizability – directly mimicking the similar actions in the training demonstrations at this position.

## 4.3 ABLATION STUDIES

In this section, we ablate the effectiveness of design choices described in Section 3. We provide additional studies in Appendix A, including training with different amounts of data, video exploration frequency $q_v$, and different horizons of the video model and goal-conditioned policy.

**Random Action Bootstrapping.** We conduct ablation studies on the importance of the random action bootstrapping technique. We first present the training-time exploration efficiency in Figure 8. As shown by *w/o rand*, the no random action method tends to collapse and can hardly achieve any task success during the exploration. Since the task completion data cannot be collected from the environment, *w/o rand* cannot obtain any meaningful results in test time, which is reflected in Table 5. In contrast, as shown by the blue, red, and orange lines in Figure 8, the model is able to obtain

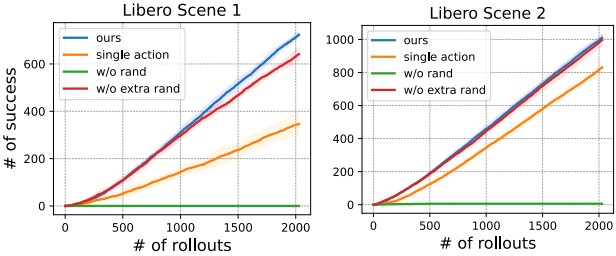

| Method | Scene 1 | Scene 2 |
|---|---|---|
| w/o rand | 0.0±0.0 | 0.0±0.0 |
| w/o video | 0.0±0.0 | 0.0±0.0 |
| w/o extra rand | 28.0±5.0 | 61.6±9.6 |
| single action | 16.4±5.4 | 41.0±5.8 |
| **Ours** | **42.2±8.9** | **62.0±8.7** |

Figure 8: **Line Chart of Number of Success versus Number of Total Exploration Rollouts During Training.** Both *ours*, *w/o extra rand*, *single action* can effectively complete tasks during video-guided exploration and thus is able to collect high-quality data for further policy training.

Table 5: **Ablation Studies on Exploration in Libero environment.** We report experiment results of w/o any random bootstrapping, w/o video-guided exploration, w/ initial but w/o extra random bootstrapping, using a single action prediction model along with our proposed approach.

significantly more task success after we apply random action bootstrapping. The performance of *w/o extra rand* decreases through time in Scene 1, showing that extra random exploration is able to stabilize training, and its performance is similar to *ours* in Scene 2, probably because the manipulation region for Scene 2 focuses on the center area of the table (see Figure 12) and the initial random actions suffice. Besides, we observe that the slope of the curve gradually becomes stable after just 250 rollout attempts, which means that the success rate of exploration can easily converge, showing the efficiency of the proposed video-guided exploration scheme.

**Chunk-level Action Prediction.** We compare the performance of the proposed chunk-level action prediction model v.s. single action prediction model. We use the same setup as BC (ResNet18+MLP) for the *single action* prediction baseline. We first present a line chart of the number of success during exploration v.s. total number of exploration rollouts in Figure 8. Both the chunk-level method and the single action method achieve non-trivial numbers of success during exploration. However, we can see that the chunk-level model consistently obtains higher success rates (i.e., steeper slope) in the exploration phase, which facilitates the policy learning because the training data contains more trajectories that successfully complete the tasks. In Table 5, we present the test-time success rate of the two methods in Libero. Similar to the exploration phase, the chunk-level action prediction model achieves higher success rates by 25.8% and 21.0%, respectively.

**Video Guided Environment Exploration.** We further compare the performance of purely random action exploration versus with video-guided exploration, as shown by *w/o video* in Table 5, where we replace the video-guided exploration in Algorithm 1 by random exploration. Unsurprisingly, the purely random exploration baseline fails across all tasks. We hypothesize that without guidance from the video model, the agent can hardly discover the necessary states to complete the tasks, probably due to the long temporal distance between the start and goal states and the fact that the relevant state space for task completion only accounts for a tiny subset of the infinite possible states.

## 5 DISCUSSION

**Limitations.** Our approach has several limitations. First, since the approach relies on goal-conditioned random exploration, for tasks that require very precise manipulation (*i.e.* stacking a block tower in millimeter precision), our random exploration procedure may not find the precise set of actions. In such settings, having a random exploration primitive (i.e. stacking a block on top of another) on which we do goal-conditioned policy learning may help us find such precise actions. In addition, purely random exploration in the physical world might pose additional requirements for the workbench, as sampled actions or the initial actions outputted by the learned goal-conditioned policy may cause undesired contacts between the robot and the external environment. This can be partially mitigated by integrating hard-coded safety constraints and is also a further direction of future work. Finally, our method requires dozens of video-guided rollouts to obtain a competent policy for a task. While resetting an environment to obtain these rollouts is easy in simulation, exploring how to autonomously reset and explore in the real world is a direction of future research.

**Conclusion.** In this paper, we have presented a self-supervised approach to ground generated videos into actions. As generative video models become increasingly more powerful, we believe that they will be increasingly useful for decision-making, providing powerful priors on how various tasks should be accomplished. As a result, the question of how we can accurately convert generated video plans to actual physical execution will become increasingly more relevant, and our approach points towards one direction to solve this question, through online interaction with the agent's environment.

ACKNOWLEDGMENTS

We thank the Center for Computation and Visualization at Brown University and Chen Sun for providing the computational resources. We thank Ronald Parr and Haotian Fu for the helpful discussion.

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

APPENDIX

# Table of Contents

# A    ADDITIONAL QUANTITATIVE RESULTS

In this section, we provide additional quantitative experiment results. In Section A.1, we compare our methods with BC trained on additional demonstration data. Next, in Section A.2, we investigate the effect of training the video model with different amounts of demonstration data. In Section A.3, we experiment our proposed method with video models of different horizons. Then, in Section A.4, we study how varying the video exploration frequency $q_v$ affects exploration performance. In Section A.5, we conduct ablation studies of using goal-conditioned policies with different horizons. Following this, in Section A.6, we evaluate the policy after different numbers of training iterations. In Section A.7, we compare our method with a reinforcement learning baseline with zero-shot reward. In Section A.8, we conduct ablation studies on the task success rate over the number of video-guided exploration rollouts. Finally, in Section A.9, we study the robustness of the resulting goal-conditioned policy learned by exploration to video models of different sampling rates.

## A.1    BC WITH MORE DATA

We provide comparison of our policy learned by unsupervised exploration and plain BC with increasing amount of training data. To train these BC baselines, We use the official demonstrations provided by Libero. For example, BC 10 means that we use 10 demonstrations per task to train the BC model (resulting in 80 demonstrations in total). The reported results are average across 5 checkpoints. Note that our policy does not need any action labels and directly learn how to ground the goals given by the video model via exploration in the environment. The video model is trained with 20 demonstrations, which only requires image sequences. That said, this is not a completely fair comparison, but we include this comparison to better illustrate the difficulty of the tasks.

We report the results in Table 6. The success rate of our method even outperforms BC trained with 50 demonstrations per task, while our policy is learned purely by exploration, indicating the efficacy of the proposed exploration method. We also observe that though the success rate of BC generally increases over more training data, BC 20 is able to slightly outperform BC 30 in Scene 1 while BC 30 slightly outperforms BC 40 in Scene 2. Since the BC model is trained on 8 tasks, the heterogeneous task structures can result in fluctuation in the success rate (the model might bias towards some specific tasks).

|  | put-red-mug-left | put-red-mug-right | put-white-mug-left | put-Y/W-mug-right | Overall |
|---|---|---|---|---|---|
| BC 10 | 2.4±3.2 | 8.8±6.9 | 23.2±12.0 | 5.6±4.1 | 10.0±6.5 |
| BC 20 | 8.8±5.3 | 15.2±7.8 | 32.0±12.9 | 21.6±12.3 | 19.4±9.6 |
| BC 30 | 9.6±4.1 | 14.4±9.0 | 28.0±9.1 | 15.2±11.1 | 16.8±8.3 |
| BC 40 | 8.0±5.1 | 11.2±5.3 | 44.8±19.8 | 21.6±12.0 | 21.4±10.6 |
| BC 50 | 12.8±8.9 | 20.0±12.9 | 40.0±12.1 | 18.4±6.0 | 22.8±10.0 |
| Ours | **38.4±15.3** | **40.8±7.8** | **51.2±3.9** | **38.4±8.6** | **42.2±8.9** |

|  | put-choc-left | put-choc-right | put-red-mug-plate | put-white-mug-plate | Overall |
|---|---|---|---|---|---|
| BC 10 | 3.2±3.0 | 4.0±5.1 | 2.4±4.8 | 6.4±4.1 | 4.0±4.2 |
| BC 20 | 19.2±9.3 | 12.8±9.3 | 7.2±5.3 | 20.0±11.3 | 14.8±8.8 |
| BC 30 | 13.6±12.0 | 31.2±15.9 | 5.6±6.0 | 16.8±10.6 | 16.8±11.1 |
| BC 40 | 13.6±5.4 | 28.0±10.4 | 7.2±6.4 | 12.8±5.3 | 15.4±6.9 |
| BC 50 | 19.2±9.3 | 24.0±10.4 | 12.8±7.8 | 24.8±4.7 | 20.2±8.0 |
| Ours | **70.4±12.8** | **79.2±3.9** | **72.8±6.4** | **25.6±11.5** | **62.0±8.7** |

Table 6: **BC with Different Amount of Data in Libero Environment.**

## A.2 Video Model with Different Amounts of Data

We investigate the effects of training the video model with different amounts of demonstration data. For the result shown in Table 1 in the main paper, we use 20 demonstrations per task to train the video model. In this section, we provide two ablation studies that use 10 demonstrations per task and 50 demonstrations per task, as shown in Table 7 below.

The performance of our method increases in both scenes as with more training data. While the performance uniformly increases in Scene 1, we observe that in some specific tasks of Scene 2, the performance slightly decreases. This might be due to the learning capacity of the underlying goal-reaching policy, since we only train one policy model for all eight tasks. In general, we believe that scaling up the data size, data diversity, and model complexity can further buttress the performance.

| | put-red-mug-left | put-red-mug-right | put-white-mug-left | put-Y/W-mug-right | Overall |
|---|---|---|---|---|---|
| train size 10 | 18.4±8.2 | 31.2±4.7 | 22.4±2.0 | 44.8±5.9 | 29.2±5.2 |
| train size 20 | 38.4±15.3 | 40.8±7.8 | 51.2±3.9 | 38.4±8.6 | 42.2±8.9 |
| train size 50 | **40.0**±10.7 | **54.4**±9.0 | **69.6**±14.9 | **57.6**±9.7 | **55.4**±11.1 |

| | put-choc-left | put-choc-right | put-red-mug-plate | put-white-mug-plate | Overall |
|---|---|---|---|---|---|
| train size 10 | 68.0±6.7 | 78.4±4.1 | 63.2±6.4 | 19.2±5.3 | 57.2±5.6 |
| train size 20 | **70.4**±12.8 | 79.2±3.9 | **72.8**±6.4 | 25.6±11.5 | 62.0±8.7 |
| train size 50 | 67.2±5.9 | **84.8**±6.4 | 67.2±6.9 | **52.0**±11.3 | **67.8**±7.6 |

Table 7: **Training the Video Model with Different Amount of Data.**

## A.3 Video Model with Different Horizons

In this section, we study the effect of different video model horizons. We follow the training procedure of the video model in Ko et al. (2023): during training, given a start image observation, we uniformly sample $h$ images between the start image and the final task completion image and use these $h$ images as supervision signals. That said, a longer video horizon $h$ will generate a denser subgoal sequence. Following Ko et al. (2023), we set horizon $h = 7$ across all our experiments except for this ablation study in Table 8.

As shown in Table 8, our method is able to maintain a similar performance across different video horizons. We observe that performance decreases when video horizon is set to 9. One potential reason is that the modeling complexity increases when we increase the prediction horizon, while we keep the same video model architecture, suggesting that we should learn video models over a sparser set of video frames.

| | put-red-mug-left | put-red-mug-right | put-white-mug-left | put-Y/W-mug-right | Overall |
|---|---|---|---|---|---|
| Video Horizon 7 | 38.4±15.3 | 40.8±7.8 | 51.2±3.9 | 38.4±8.6 | 42.2±8.9 |
| Video Horizon 8 | 28.8±3.0 | 59.2±5.3 | 52.0±8.8 | 54.4±9.3 | 48.6±6.6 |
| Video Horizon 9 | 17.6±5.4 | 65.6±9.7 | 38.4±8.6 | 30.4±3.2 | 38.0±6.7 |

| | put-choc-left | put-choc-right | put-red-mug-plate | put-white-mug-plate | Overall |
|---|---|---|---|---|---|
| Video Horizon 7 | 70.4±12.8 | 79.2±3.9 | 72.8±6.4 | 25.6±11.5 | 62.0±8.7 |
| Video Horizon 8 | 46.4±6.5 | 69.6±10.9 | 64.0±4.4 | 31.2±8.5 | 52.8±7.6 |
| Video Horizon 9 | 52.0±6.7 | 70.4±3.2 | 68.8±6.9 | 13.6±7.8 | 51.2±6.2 |

Table 8: **Video Model with Different Horizons.**

## A.4 DIFFERENT VIDEO EXPLORATION FREQUENCY

In this section, we study the effect of video exploration frequency. For example, when $q_v$ is set to 200, we conduct one video-guided exploration for each task every 200 training iterations. The trade-off between smaller and larger $q_v$ is that: if $q_v$ is small, the agent will conduct exploration in a higher frequency, where the replay buffer will refresh faster and contain more latest rollout data. If $q_v$ is large, the agent conducts exploration less frequently, which might enable the agent to better fit and digest the existing data in the replay buffer. We provide ablation studies on 5 different $q_v$ on Libero environment, as shown in Figure 9.

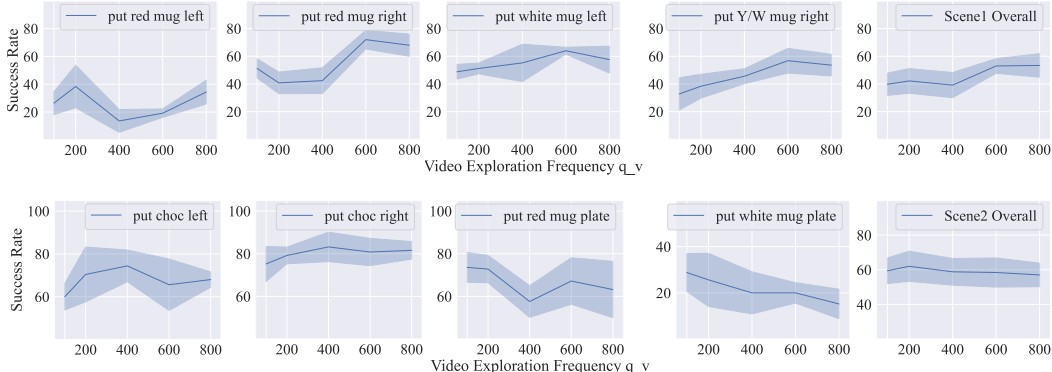

Figure 9: **Different Video Exploration Frequency** $q_v$. $q_v = 200$ indicates that we conduct one video-guided exploration for each task every 200 training iterations. Thus, the agent will conduct more exploration with lower $q_v$ while less exploration with higher $q_v$. Our method performs steadily across different values of $q_v$.

Figure 9 shows that our method performs steadily across various $q_v$, ranging from 100 to 800. Note that when $q_v$ is set to 800, the video exploration is four times smaller than our default setting ($q_v = 200$), which means that our method can achieve a comparable performance with much fewer environment interactions.

## A.5 DIFFERENT POLICY HORIZON FOR OUR GOAL CONDITIONED POLICY

In this section, we study the effect of using different horizons for the diffusion policy based goal-conditioned policy. We set the policy horizons to 12, 16, 20, and 24 with the same CNN-based architecture. We observe that the policy learned by the proposed exploration pipeline is robust to the different horizons value, and obtain the highest success rate when horizon = 16, which also conforms to the reported results in Diffusion Policy (Chi et al., 2023). Please refer to Table 9 for quantitative results.

|  | put-red-mug-left | put-red-mug-right | put-white-mug-left | put-Y/W-mug-right | Overall |
|---|---|---|---|---|---|
| Horizon=12 | 21.6±7.4 | **56.8**±4.7 | 43.2±9.9 | 47.2±6.9 | **42.2**±7.2 |
| Horizon=16 | **38.4**±15.3 | 40.8±7.8 | **51.2**±3.9 | 38.4±8.6 | **42.2**±8.9 |
| Horizon=20 | 13.6±6.0 | 50.4±7.8 | 48.0±7.2 | 37.6±7.0 | 37.4±7.0 |
| Horizon=24 | 21.6±6.0 | 36.8±4.7 | 40.8±8.5 | **48.8**±8.5 | 37.0±6.9 |

|  | put-choc-left | put-choc-right | put-red-mug-plate | put-white-mug-plate | Overall |
|---|---|---|---|---|---|
| Horizon=12 | 62.4±10.3 | **80.0**±7.2 | 56.0±7.2 | **25.6**±8.6 | 56.0±8.3 |
| Horizon=16 | **70.4**±12.8 | 79.2±3.9 | **72.8**±6.4 | **25.6**±11.5 | **62.0**±8.7 |
| Horizon=20 | 64.8±14.2 | 76.0±5.1 | 68.8±15.3 | 18.4±6.0 | 57.0±10.1 |
| Horizon=24 | 60.0±12.1 | 68.8±10.6 | 37.6±13.8 | 20.8±7.3 | 46.8±10.9 |

Table 9: **Different Horizons for the Goal-Conditioned Policy.**

## A.6 DIFFERENT TRAINING TIMESTEPS

In this section, we investigate the model performance at different training timesteps on Libero environment, specifically, when the model is trained for $40k$, $80k$, $120k$, $160k$, and $200k$ steps. We set the video exploration frequency $q_v$ to 200 in this experiment. We observe that the overall success rate of the two scenes stabilize after $80k$ training steps. However, the per-task success rate oscillates through further training. This might be caused by the recent collected data in the replay buffer (which is used for training) and the task level interference.

| | put-red-mug-left | put-red-mug-right | put-white-mug-left | put-Y/W-mug-right | Overall |
|---|---|---|---|---|---|
| $40k$ | 34.4±7.4 | 37.6±12.0 | 28.8±4.7 | 32.0±10.4 | 33.2±8.6 |
| $80k$ | **63.2**±9.3 | **72.0**±6.7 | 48.8±6.9 | **42.4**±4.1 | **56.6**±6.7 |
| $120k$ | 47.2±8.5 | 56.0±4.4 | **58.4**±9.3 | 38.4±4.8 | 50.0±6.8 |
| $160k$ | 33.6±7.4 | 43.2±5.9 | 53.6±6.5 | 33.6±10.9 | 41.0±7.7 |
| $200k$ | 38.4±15.3 | 40.8±7.8 | 51.2±3.9 | 38.4±8.6 | 42.2±8.9 |

| | put-choc-left | put-choc-right | put-red-mug-plate | put-white-mug-plate | Overall |
|---|---|---|---|---|---|
| $40k$ | 13.6±9.0 | 14.4±6.0 | 9.6±6.0 | 5.6±2.0 | 10.8±5.7 |
| $80k$ | 47.2±6.9 | 58.4±8.6 | 68.8±6.9 | 15.2±5.9 | 47.4±7.1 |
| $120k$ | 51.2±3.0 | 70.4±8.6 | 68.0±6.7 | 23.2±8.2 | 53.2±6.6 |
| $160k$ | 66.4±8.6 | 74.4±5.4 | **75.2**±5.3 | 16.0±8.0 | 58.0±6.8 |
| $200k$ | **70.4**±12.8 | **79.2**±3.9 | 72.8±6.4 | **25.6**±11.5 | **62.0**±8.7 |

Table 10: **Performance at Different Training Steps in Libero Environment.**

## A.7 COMPARISON TO RL WITH ZERO-SHOT REWARD

In this section, we compare our method with a reinforcement learning baseline. Since our method does not have access to environment rewards, we leverage a foundational zero-shot robotic model to generate the rewards. Specifically, we adopt DrQ (Kostrikov et al., 2020; Yarats et al., 2021) and LIV (Ma et al., 2023) as the RL method and foundation model respectively. We use the potential-based reward defined in LIV.

We present the results in Table 11. We see that DrQ+LIV fails to make meaningful progress in five out of six tasks, achieving only an 8% success rate in the handle-press task. We observe that one probable cause is that the generated reward signals are ambiguous, which hinders the RL method to learn. For instance, it is challenging for the model to generate a significant positive reward upon task completion and the rewards may oscillate even when positive progress is being made.

| | door-open | door-close | handle-press | hammer | assembly | faucet-open | Overall |
|---|---|---|---|---|---|---|---|
| DrQ + LIV | 0.0±0.0 | 0.0±0.0 | 8.0±0.0 | 0.0±0.0 | 0.0±0.0 | 0.0±0.0 | 1.3±0.0 |
| Ours | **76.0**±4.9 | 85.6±2.0 | **94.4**±2.0 | **43.2**±6.4 | **16.8**±3.0 | **87.2**±3.0 | 67.2±3.6 |

Table 11: **Quantitative Comparison to DrQ+LIV on 6 tasks from Meta-World.**

### A.8 Success Rate v.s. Number of Video-Guided Rollouts

In this section, we conduct ablation studies on the task success rate over the number of video-guided rollouts. We show the performance curve of the Libero task *put the red mug on the plate* in Figure 10, where the y-axis represents the success rate of 25 evaluation episodes and the x-axis denotes the number of conducted video-guided exploration rollouts.

Specifically, we first warm-start the policy by training it on randomly sampled actions, which corresponds to zero video-guided rollouts. Then, the model begins to perform video-guided exploration, and we set the video exploration frequency to 800 (i.e., the agent will conduct 5 video-guided exploration every 800 training iterations). We save a model checkpoint after every 10 video exploration episodes and evaluate each checkpoint on 25 test-time problems. Note that to better demonstrate the correlation of success rate and # of video-guided exploration, we train the policy model just on a single task, while in Table 1, the policy model is jointly trained on 8 tasks.

As shown in Figure 10, the policy is unable to accomplish the task without video-guided exploration episodes (when the x-axis is 0). However, the success rate rapidly increases with the start of video-guided exploration episodes, reaching a success rate of 40% after only 100 episodes. After approximately 200 episodes, the success rate gradually saturates, stabilizing around 80%.

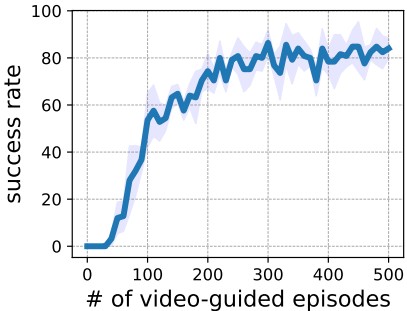

Figure 10: **Performance Curve of task *put the red mug on the plate* in Libero.** The y-axis represents the success rate over 25 unseen test-time problems and the x-axis denotes the number of conducted video-guided exploration episodes. The task success rate increases rapidly with the start of video-guided exploration, while the agent struggles to complete the task without it.

A.9   POLICY ROBUSTNESS TO VIDEO MODELS OF DIFFERENT SAMPLING RATES

In this section, we study the robustness of the resulting goal-conditioned policy learned by exploration to video models of different sampling rates. Specifically, once the policy is trained, we replace the training-time video model with video models of different sampling rates while remain using the same policy.

Following the video model design in Ko et al. (2023), we use the *Number of Frames to Goal* to denote the sampling rate of the video model. For example, *Number of Frames to Goal = 9* indicates that the video model is designed to uniformly generate 9 frames from the initial observation to the goal state, that is, compared to a 7-frame video model, the sampling rate is denser (i.e., the temporal distance between two adjacent frames is smaller).

We present the policy success rate in Table 12. In this table, the policy is trained with a 7-frame video model (marked with an asterisk) while the number of frames to goal of the test-time video model varies from 5 to 9. The policy consistently achieves high performance across video models with different sampling rates, demonstrating its robustness to variations in the sampling rate during testing. This is probably because of the robust policy training enabled by the combination of random action bootstrapping and video-guided exploration.

We observe that the 6-frame video model attains the highest success rate, even surpassing the training-time 7-frame model. This improvement is likely due to the better quality of the synthesized video frames, as modeling a greater or denser number of frames increases complexity while we keep the video model size unchanged in this experiment.

| # Frames to Goal | put-red-mug-left | put-red-mug-right | put-white-mug-left | put-Y/W-mug-right | Overall |
|---|---|---|---|---|---|
| 5 | 32.8±12.0 | 46.4±10.3 | 66.4±5.4 | **41.6**±4.8 | 46.8±8.1 |
| 6 | 38.4±4.8 | 48.0±8.8 | **68.8**±4.7 | 36.0±6.7 | **47.8**±6.2 |
| 7* | 38.4±15.3 | 40.8±7.8 | 51.2±3.9 | 38.4±8.6 | 42.2±8.9 |
| 8 | **39.2**±6.4 | **51.2**±13.7 | 65.6±4.8 | 25.6±5.4 | 45.4±7.6 |
| 9 | 32.8±5.3 | 44.0±8.8 | 36.8±9.9 | 28.8±5.9 | 35.6±7.5 |

| # Frames to Goal | put-choc-left | put-choc-right | put-red-mug-plate | put-white-mug-plate | Overall |
|---|---|---|---|---|---|
| 5 | 62.4±4.1 | 76.8±11.7 | 66.4±4.1 | **59.2**±10.6 | 66.2±7.6 |
| 6 | 64.8±6.9 | 72.0±12.4 | **74.4**±9.3 | 58.4±9.3 | **67.4**±9.5 |
| 7* | **70.4**±12.8 | 79.2±3.9 | 72.8±6.4 | 25.6±11.5 | 62.0±8.7 |
| 8 | 52.0±12.1 | **84.8**±8.2 | 68.0±8.4 | 26.4±10.9 | 57.8±9.9 |
| 9 | 59.2±13.9 | 76.0±8.8 | **74.4**±9.7 | 20.8±7.3 | 57.6±9.9 |

Table 12: **Policy Performance with Video Models of Different Sampling Rates at Test Time.** All results in the table are from the same goal-conditioned policy but with video models of different sampling rates (identified by the number of intermediate frames from initial observation to the goal state). The policy uses a video model of *# Frames to Goal = 7* to conduct video-guided exploration during training, but is evaluated with video models of different numbers of to-goal frames at test time. The policy performs consistently and is robust to various sampling rates at test time.

B   ADDITIONAL QUALITATIVE RESULTS

In this section, we present additional qualitative results in both tabletop manipulation environments and visual navigation environments. For extra results and side-by-side qualitative comparison, please refer to our website.

In Section B.1, we provide qualitative results for each task in Libero environment. In Section B.2, we show qualitative results for Meta-World tasks. Next, in Section B.3, we present qualitative results on Calvin environment. Finally, in Section B.4, we present a long-horizon visual navigation planning rollout in iTHOR environment.

B.1   LIBERO

In this section, we provide additional qualitative results of our method on 8 tasks in Libero environment, as shown in Figure 11 and 12. For each episode, we show the generated video in the first row

and the rollout results of our goal conditioned policy learned by self-supervised exploration in the second row.

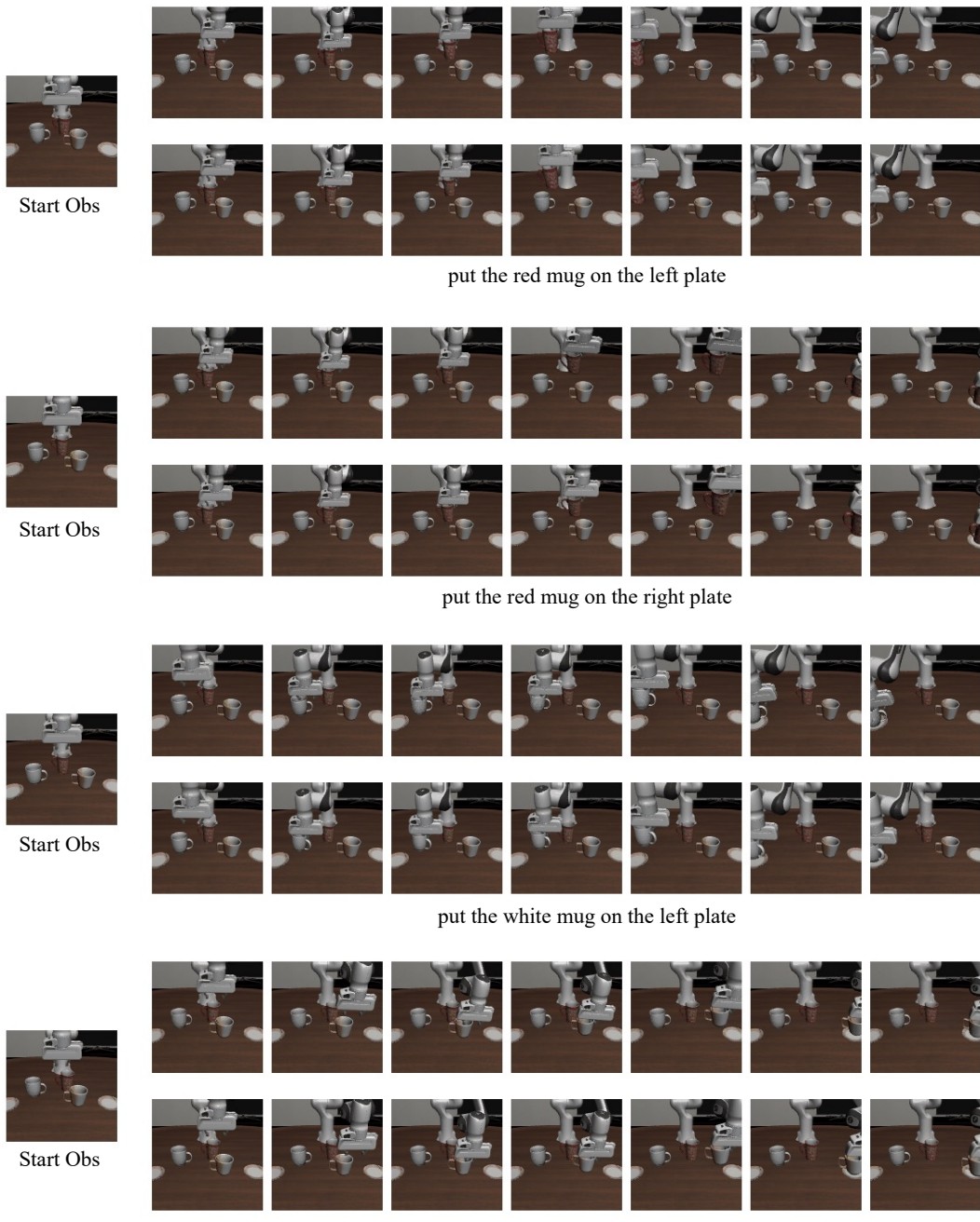

Figure 11: **Additional Qualitative Results of the Generated Videos and Our Policy Rollout on Libero Scene 1.** We present qualitative results of four tasks in Libero Scene 1. For each task, results are displayed in 2 rows: the first row of images are the generated video from the video model conditioned on the start observation image and task description, and the second row of images are the rollout of our policy conditioned on the corresponding frames from the generated video. The task description provided to the video model are shown below the respective images.

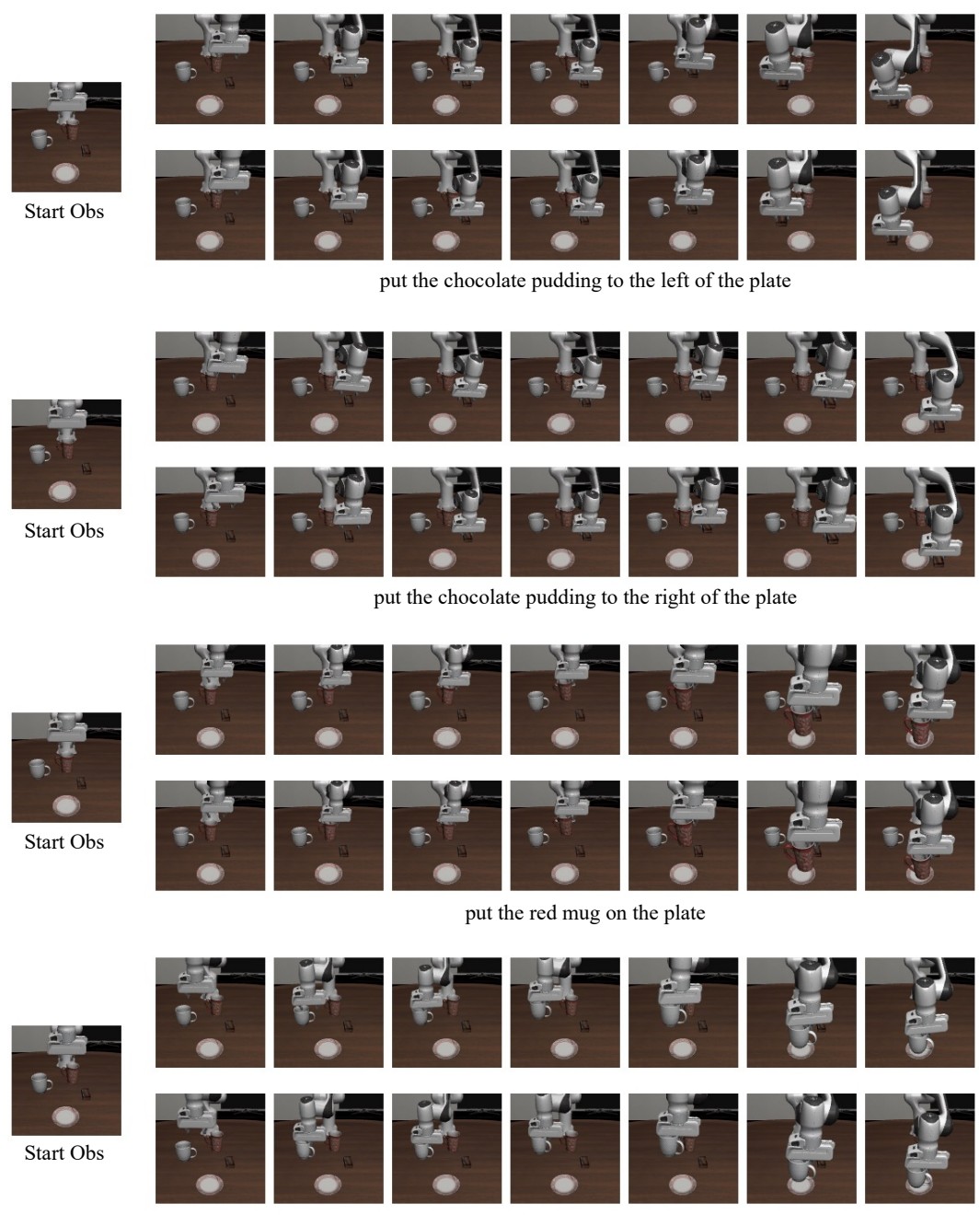

Figure 12: **Additional Qualitative Results of the Generated Videos and Our Policy Rollout on Libero Scene 2.** We present qualitative results of four tasks in Libero Scene 2. For each task, results are displayed in 2 rows: the first row of images are the generated video from the video model conditioned on the start observation image and task description, and the second row of images are the rollout of our policy conditioned on the corresponding frames from the generated video. The task description provided to the video model are shown below the respective images.

## B.2 META-WOLRD

In this section, we provide additional qualitative results of our method on 6 tasks in Meta-World environment, as shown in Figure 13 and 14. For each episode, we show the generated video in the

first row, and the rollout results of our goal-conditioned policy learned by self-supervised exploration in the second row.

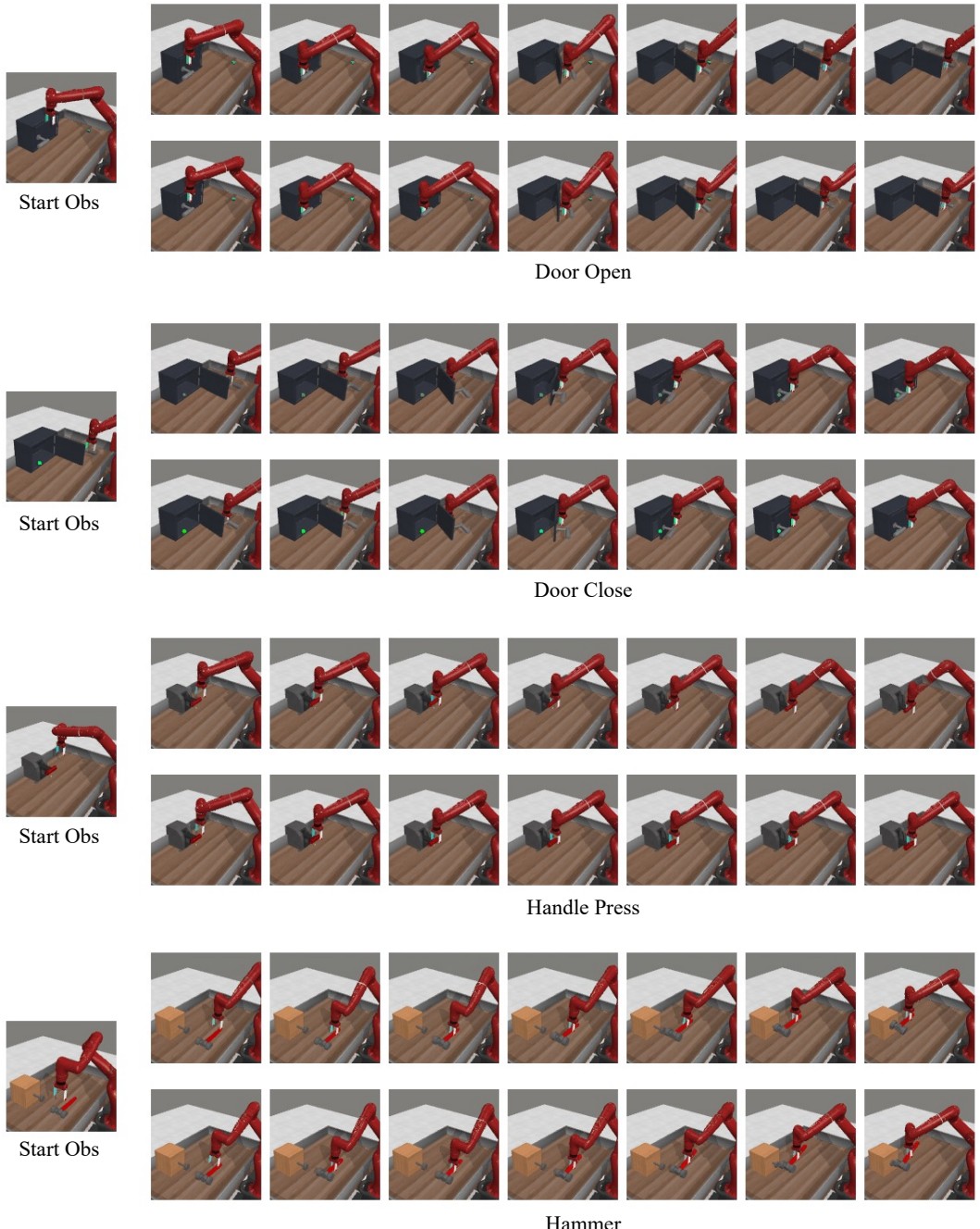

Figure 13: **Additional Qualitative Results of the Generated Videos and Our Policy Rollout of 4 tasks on Meta-World.** We present qualitative results of four tasks in Meta-World Environment. For each task, results are displayed in 2 rows: the first row of images are the generated video from the video model conditioned on the start observation image and task description, and the second row of images are the rollout of our policy conditioned on the corresponding frames from the generated video. The task prompts provided to the video model are shown below the respective images.

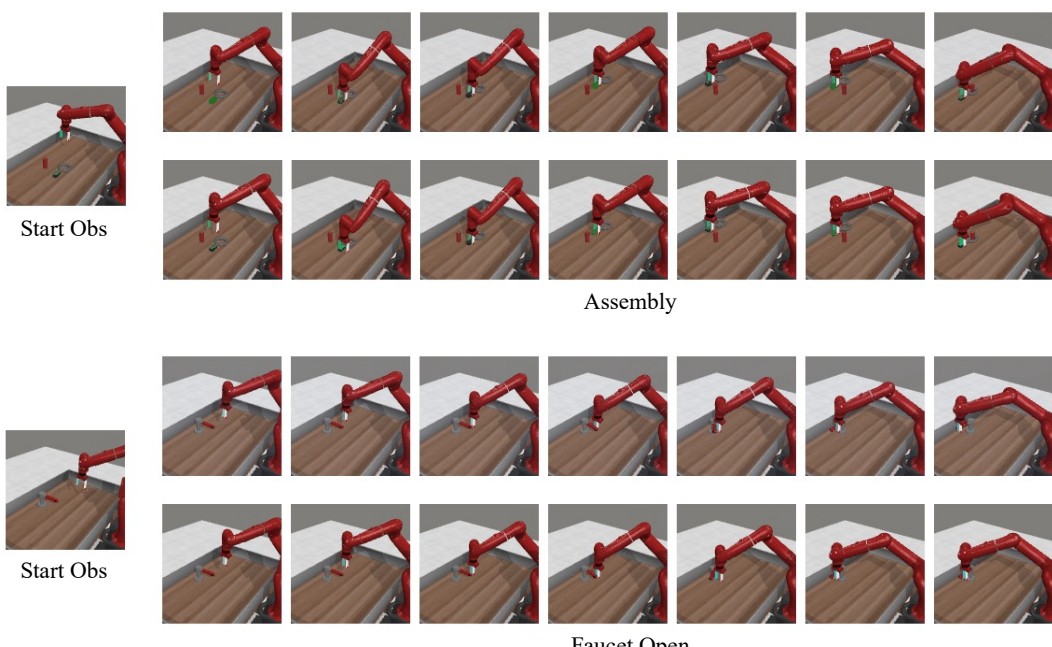

Figure 14: **Additional Qualitative Results of the Generated Videos and Our Policy Rollout of 2 tasks on Meta-World.** We present qualitative results of four tasks in Meta-World Environment. For each task, results are displayed in 2 rows: the first row of images are the generated video from the video model conditioned on the start observation image and task description, and the second row of images are the rollout of our policy conditioned on the corresponding frames from the generated video. The task prompt provided to the video model are shown below the respective images.

B.3 CALVIN

In this section, we provide additional qualitative results of our method on 4 tasks in Calvin environment, as shown in Figure 15. For each episode, we show the generated video in the first row, and the rollout results of our goal conditioned policy learned by self-supervised exploration in the second row.

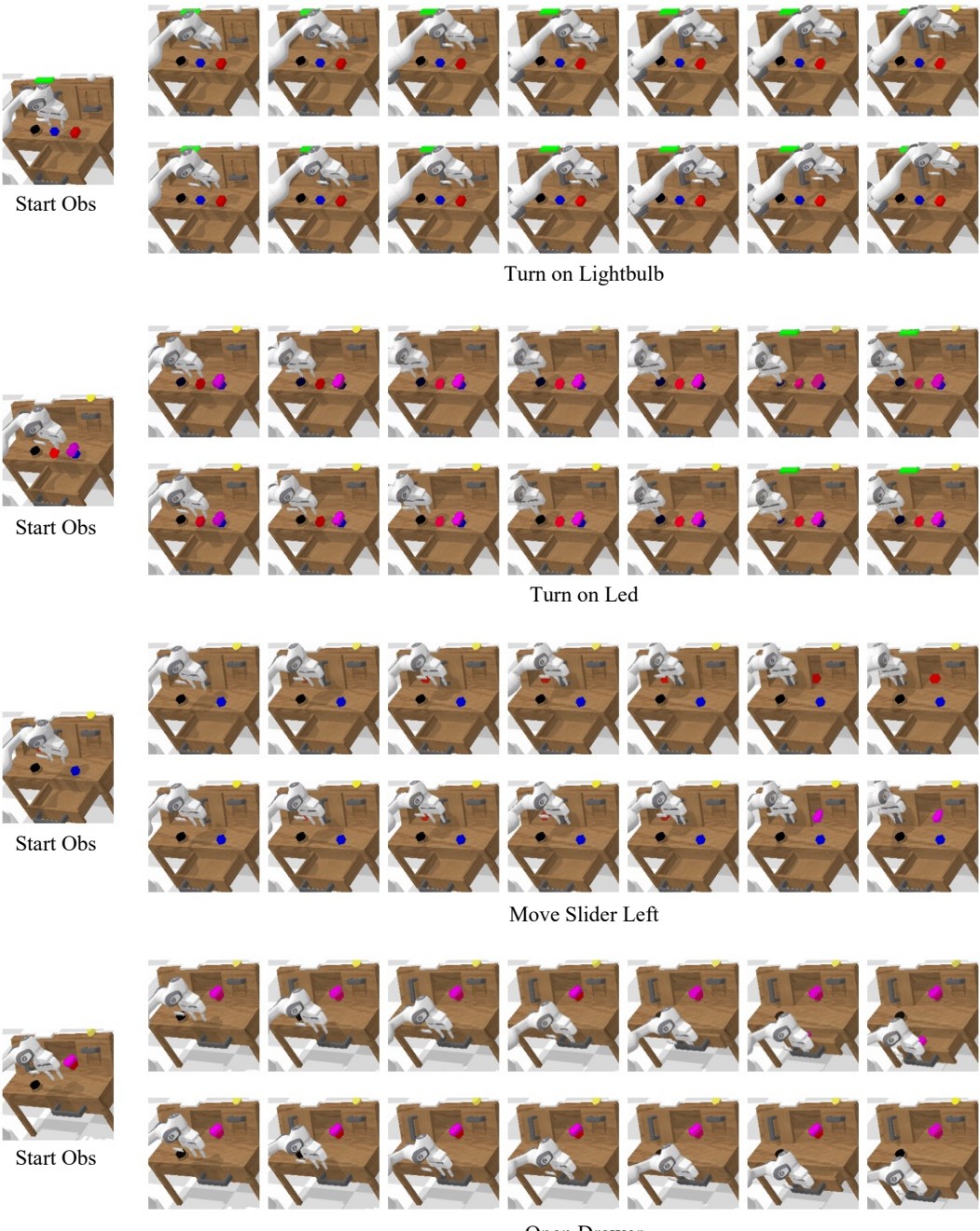

Figure 15: **Additional Qualitative Results of the Generated Videos and Our Policy Rollout of 4 tasks on Calvin.** We present qualitative results of four tasks in Calvin Environment. For each task, results are displayed in 2 rows: the first row of images are the generated video from the video model conditioned on the start observation image and task description, and the second row of images are the rollout of our policy conditioned on the corresponding frames from the generated video. The task prompt provided to the video model are shown below the respective images.

B.4    iThor Visual Navigation

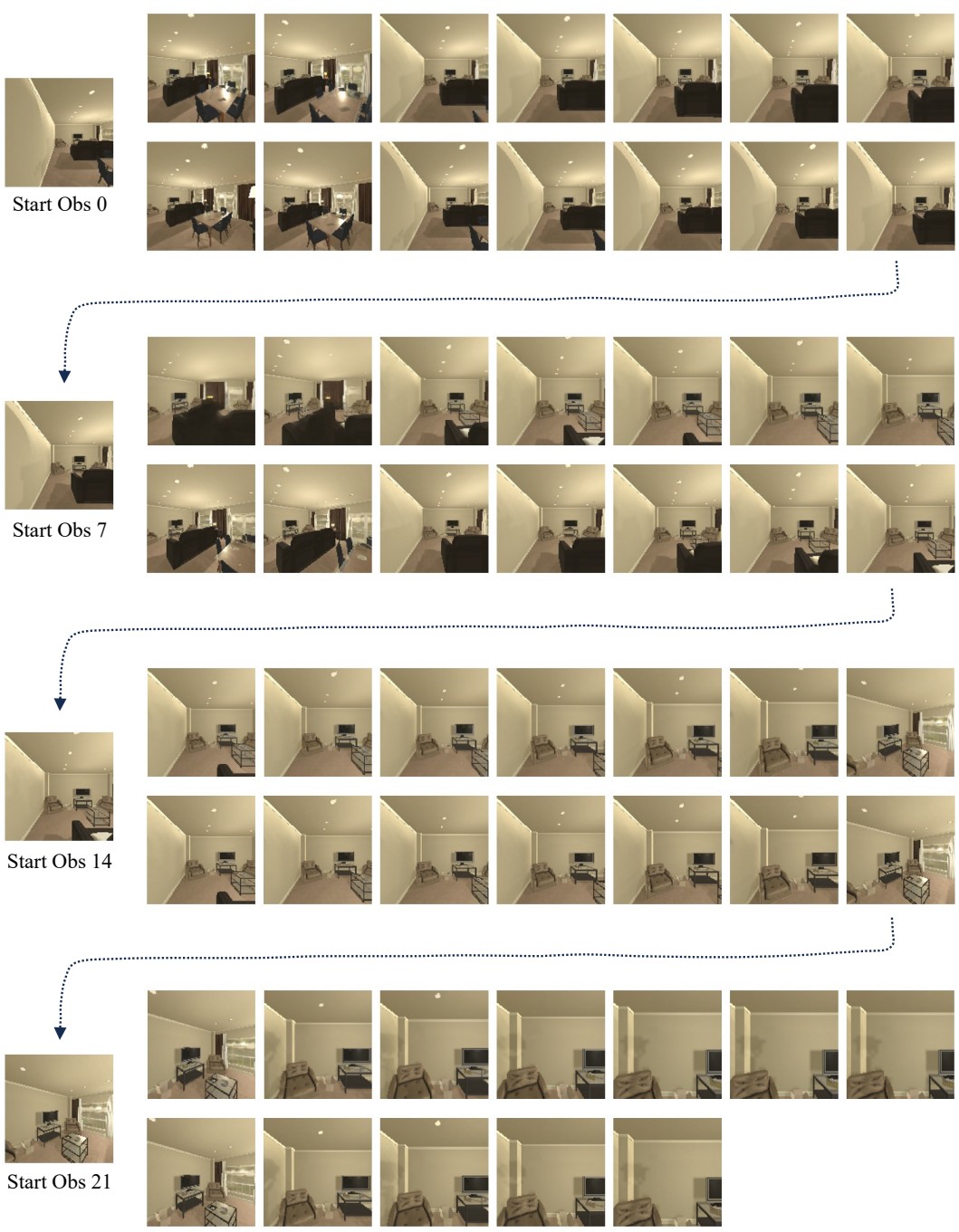

iTHOR Living Room FloorPlan201-- Television

Figure 16: **Four Consecutive Policy Rollout for one Evaluation Episode in a Living Room Scene of iTHOR.** In each rollout, the first row of images are the generated video from the video model conditioned on the start observation image (shown in the leftmost column) and task description (name of the target). The second row of images are the rollout of our policy conditioned on the corresponding frames from the generated video. The names of the scene and target are shown at the bottom. Due to the long spatial distance, the agent takes more than 20 actions to navigate to the television, which corresponds to the four videos above. The last observation of the previous rollout is fed to the video model to generate future subgoals, as indicated by the dashed arrows.

In this section, we provide additional qualitative results of the generated videos and our goal conditioned policy learned by self-supervised exploration in a Living Room Scene (FloorPlan201) of iTHOR visual navigation environments, as shown in Figure 16. Starting at *Start Obs 0*, the agent is tasked to navigate to the television at the other end of the room, which typically requires more than 20 actions to reach. While the horizon of the video model is only 7, we consecutively generate future subgoals by feeding the last observation image to the video model as condition (indicated by the dashed arrows). With four video predictions and 26 actions, the agent successfully navigates to the television, showing the long and consecutive rollout capability of our policy.

## C  FAILURE MODE ANALYSIS

In the previous section, we demonstrate that our method is able to achieve non-trivial performance in various robotic environments. However, the method still poses some limitations, which lead to failure. In this section, we provide analysis into the failure mode of our method. We categorize the causes to video model based and policy learning based.

### C.1  VIDEO MODEL

The performance of the video model is one influential factor of the success rate because that if an incorrect subgoal is given, even though the policy is accurate enough, the task cannot be completed.

**Hallucination** is a common issue for generative models (Yang et al., 2024a; Ji et al., 2023). We also observe some extent of hallucination in our video model. We present some visualizations in Figure 17.

In Table 1, *Ours* and *Ours w/ SuSIE* only achieve 25.6% and 36.0% on the task 'put the white mug on the plate'. We observe that one major cause of this relatively low performance is the heavy hallucination in the generated video of this task, as shown in the first row of Figure 17. We hypothesize that this might partially due to the dataset imbalance problem, as we have two tasks involving chocolate pudding while only one for white mug in this scene, considering that only 20 demonstrations are provided for each task.

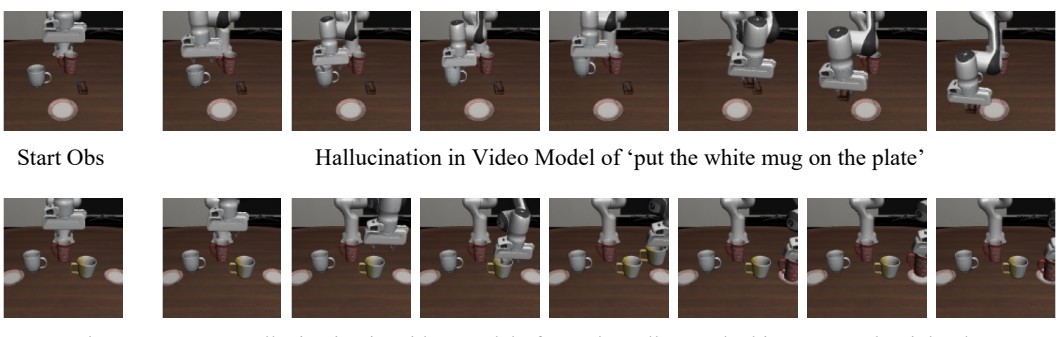

Start Obs             Hallucination in Video Model of 'put the white mug on the plate'

Start Obs          Hallucination in Video Model of 'put the yellow and white mug on the right plate'

Figure 17: **Failure Mode: Video Model Hallucination.** In the first row, the given prompt to the video model is 'put the white mug on the plate'. Though the first four frames generated by the video model are on the right path for achieving the goal, the white mug is suddenly replaced by the chocolate pudding in the following frames, which will subsequently confuse the policy. In the second row, the given prompt to the video model is 'put the yellow and white mug on the right plate'. Similarly, the agent in the video frames first correctly approaches the yellow and white mug, but the yellow and white mug is replaced by a red mug in the intermediate frames.

**Mismatch of Task and Generated Videos** Given a specific task description, we observe that the video model might generate a video that is actually for completing another task. For example, given the task description of 'put the chocolate pudding to the *left* of the plate', the video model might generate a video that 'put the chocolate pudding to the *right* of the plate'. One potential cause is the relatively close embedding distance between these two sentences as well as the corresponding videos, making the model generate a similar but incorrect video. We provide some visualizations in Figure 18.

We believe that these issue can be mitigated by multiple ways, such as improving the video model architecture, scaling up the training data, finetuning from existing pre-trained large video model, or using the environment interaction feedback to correct the video model, which might be interesting future research directions.

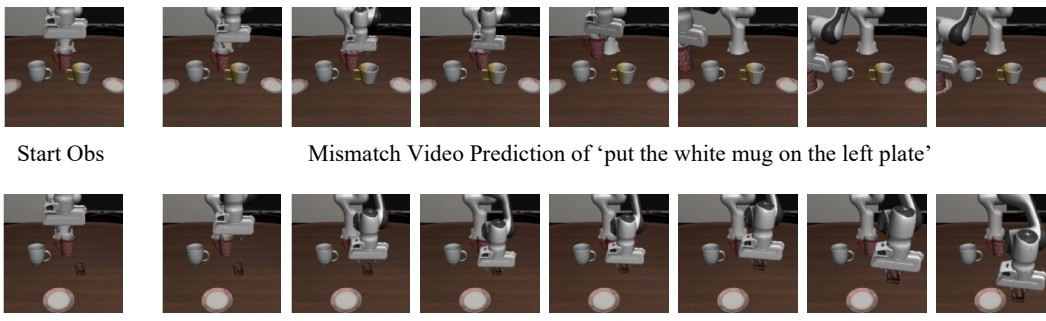

Figure 18: **Failure Mode: Mismatch of the Task and the Generated Video.** In the first row, the given prompt to the video model is 'put the white mug on the left plate', while the generated video puts the red cup to the left plate. In the second row, the given prompt to the video model is 'put the chocolate pudding to the left of the plate', while the generated video puts the chocolate pudding to the right of the plate.

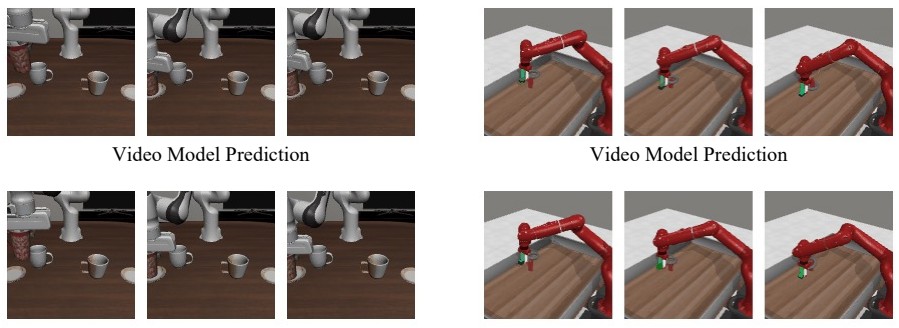

Figure 19: **Failure Mode: Precision, Libero.** A failure case of 'put the red mug on the left plate' in Libero Scene 1. We only show the results of the last three subgoals. Though the goal conditioned policy is able to successfully put the red mug on the left plate in the last frame, a slight displacement is introduced (see the rightmost column), which results in task failure.

Figure 20: **Failure Mode: Precision, Meta-World.** A failure case of the assembly task. We only show the results of the last three subgoals. In this episode, the goal conditioned policy exactly follows the subgoals in the first two frames, but in the last frame (the rightmost column), the nut touches the peg due to slight precision error and hence fails to insert the nut into to peg, resulting in task failure.

## C.2 POLICY LEARNING

Policy learning is particularly challenging in such unsupervised setting since we assume no action demonstrations nor rewards. Generally, we observe that the learned policy is able to follow the video model, but there remain challenges for the policy to handle fine-grained tasks that require precise control. For example, our policy achieves 94.4% at handle press task, while only achieves 16.8% at assembly task in Meta-World. In Figure 19 and Figure 20, we provide some qualitative results of failure cases in Libero and Meta-World.

In Figure 19, we present a failure case of the task 'put the red mug on the left plate' in Libero. In this task, the agent needs to put the red mug at the center of the left plate within a small tolerance. One major reason of the policy's failure is that the mug is not precisely placed at the center, as shown in the rightmost column.

In Figure 20, we present a failure case of the task assembly in Meta-World. In this task, the agent needs to pick up a nut and precisely place it onto a peg. While the policy seems to exactly follow the given subgoals, it fails to insert the nut at the last frame where the nut undesirably touches the peg.

One potential way to increase the performance is to design some replanning strategies. Another interesting direction to improve the policy is to better leverage the geometric information. Currently, our policy only takes one 2D observation image as input. Using a complementary 3D point cloud for observation input is likely to be beneficial (Ze et al., 2024; Ling et al., 2023) for fine-grained tasks. In addition, incorporating some action primitive in the random exploration phase, learning or defining some manipulation primitives for precise control task might also be helpful.

## D  IMPLEMENTATION DETAILS

In this section, we describe the implementation of our proposed method and baselines.

**Software:** The computation platform is installed with Red Hat 7.9, Python 3.9, PyTorch 2.0, and Cuda 11.8

**Hardware:** For each of our experiments, we used 1 NVIDIA RTX 3090 GPU or a GPU of similar configuration.

### D.1  MODEL ARCHITECTURES

For the architecture of the video model, we adopt the same design from AVDC (Ko et al., 2023), which uses 3D UNet as video denoiser and CLIP to encode language features. In tabletop manipulation tasks, We instantiate the goal-conditioned policy by a diffusion policy (Chi et al., 2023), which uses ResNet18 as image encoder and 1D UNet to denoise the action trajectory. Following the default setup in (Chi et al., 2023), we set the horizon to 16 across all other experiments. Finally, for iTHOR environment, since the action space is discrete and the required number of actions to reach goal is smaller, we use a ResNet18 with a 3-layer MLP with ReLU activation as the goal-conditioned policy with an output horizon of 1.

### D.2  TRAINING DETAILS

**Training Pipeline** In training, we randomly sample sequences of image-action pairs of horizon $h$ from the replay buffer. For Diffusion Policy setup, we mainly follow the hyperparameters suggested in the original paper. We provide detailed hyperparameters for training our model in Table 13 and 14. We do not apply any hyperparameter search or learning rate scheduler. For Libero environment, the training time of our model is approximately 36 hours for a full 200k training steps. However, the performance can gradually saturate with much fewer steps. For the Meta-World and iThor environments, the training time of our model is approximately one day; for Calvin, the training time of our model is around 15 hours. For these environments, we also observe that the performance can gradually saturate within less training time.

**Exploration Grasping Primitive.** To improve the exploration efficiency of grasping, we further add a simple grasping primitive that encourages the agent to attempt a grasp when the end effector is positioned closely above an object. Specifically, we measure the distance from the wrist camera to the nearest obstacle and compare the distance to the current height of the gripper above table. The agent is prompted to grasp if the difference is high (indicating a nearby object). This heuristic can be applied across various robots and greatly reduces the search space necessary to learn grasping.

### D.3  RANDOM ACTION BOOTSTRAPPING DETAILS

**Random Action Bootstrapping.** We provide detailed hyperparameters for random action bootstrapping in Table 15, where we use the same notations as in Section 3 and Algorithm 1.

In Table 15, # of Initial Episodes $n_r$ and # of Addition Episodes $n'_r$ are with respect to one task for tabletop manipulation and with respect to one scene in iTHOR navigation. In Libero, the policy learns all 8 tasks concurrently; In Meta-World, the policy learns each task separately; In Calvin, the policy learns all 4 tasks concurrently; In iTHOR, the policy learns all 4 scenes (12 tasks) concurrently.

| Hyperparameters | Value |
|---|---|
| Action Prediction Horizon | 16 |
| Action Horizon | 8 |
| Diffusion Time Step | 100 |
| Iterations | 200K |
| Batch Size | 64 |
| Optimizer | Adam |
| Learning Rate | 1e-4 |
| Input Image Resolution | (128,128) |

Table 13: Hyperparameters of our Goal-conditioned Policy in Libero, Meta-World, and Calvin.

| Hyperparameters | Value |
|---|---|
| Horizon | 1 |
| Image Encoder | ResNet18 |
| MLP size | [(1024, 256), (256, 128), (128, 4)] |
| Activation | ReLU |
| Iterations | 100K |
| Batch Size | 64 |
| Optimizer | Adam |
| Learning Rate | 1e-4 |

Table 14: Hyperparameters of our Goal-conditioned Policy in iTHOR.

In addition, before video-guided exploration, we first warm-start the policy by training it solely on the initial random action episodes for $N_r$ steps.

We use the *first in, first Out* convention to implement the replay buffer. The replay buffer is shared across tasks if doing multi-task exploration, thus the replay buffer size is relatively larger for multi-task setting.

| | Libero | Meta-World | Calvin | iTHOR |
|---|---|---|---|---|
| Warm-start steps $N_r$ | 10k | 10k | 10k | 10k |
| Episode Length | 120 | 120 | 120 | 50 |
| Action Chunk Size $l_c$ | 24 | 24 | 24 | 1 |
| # of Initial Episodes $n_r$ | 50 | 200 | 100 | 100 |
| Periodic Frequency $q_r$ | 500 | 500 | 500 | 500 |
| # of Addition Episodes $n_r'$ | 2 | 10 | 4 | 2 |
| Replay Buffer Size | 1200 | 500 | 1200 | 1200 |

Table 15: Hyperparameters for Random Action Bootstrapping in each Environment. Same notations are used as in Section 3 and Algorithm 1. $n_r$ and $n_r'$ represent number of random action exploration per task.

In Table 16, we provide the values of $a_{\text{low}}$ and $a_{\text{high}}$ for sampling random actions, which are introduced in Section 3.2. For visual navigation tasks, since the action space is discrete, we sample from the four possible actions {Move Ahead, Turn Left, Turn Right, Done} with uniform probability during the random action bootstrapping.

| | Libero | Meta-World | Calvin |
|---|---|---|---|
| $a_{\text{low}}$ | [-1,-1,-1,-0.01,-0.01,-0.01] | [-1,-1,-1] | [-1,-1,-1,-1,-1,-1,-1] |
| $a_{\text{high}}$ | [1,1,1,0.01,0.01,0.01] | [1,1,1] | [1,1,1,1,1,1] |

Table 16: Values of $a_{\text{low}}$ and $a_{\text{high}}$ for Random Action Sampling. The gripper action is discretized to -1 and 1 (close and open) hence not included in the table. In Libero environment, a smaller value in the orientation action space is set for more efficient random action bootstrapping.

## D.4  TASK SUCCESS METRIC

In this section, we describe the metric to evaluate whether a task is successfully completed. Specifically, the simulation environment will check the state of the target object in manipulation tasks or check the agent state in visual navigation tasks after executing an action. For manipulation tasks, if the target object is within the range of the specified goal state, the environment will return a success; for navigation tasks, if the target object is within a certain distance to and in the view of the agent and the agent executes a 'Done' action, the environment will return a success.

During a rollout, if a success is returned, this rollout will be counted as success and terminated; otherwise, if we have finished a rollout (i.e., have sequentially executed all the synthesized video frames) and no success is returned, this episode will be counted as a failed rollout.

## D.5  IMPLEMENTATION OF BASELINES

**Behavior Cloning (BC).** For BC, the observed image is first fed to a ResNet-18 (He et al., 2016) to encode visual information. The vision feature vector is then fed into a 3-layer MLP to predict the next action. For multi-task BC, we concatenate the vision feature with task language description feature encoded by CLIP, and feed the concatenated feature to a 3-layer MLP to predict an action. We use MSE as loss and train for 100k steps with a batch size of 64.

**Diffusion Policy Behavior Cloning (DP BC).** For DP BC, following Chi et al. (2023), we use a 1D convolutional neural network as trajectory denoiser and use a ResNet-18 to encode the image observation. We set the diffusion denoising timestep to 100, horizon to 16, and train for 200k iterations with a batch size of 64.

**Goal-conditioned Behavior Cloning (GCBC).** For GCBC, we concatenate the start image and goal image and feed it to ResNet-18 to encode the visual observation. Similarly, the visual feature is then fed to a 3-layer MLP to predict the next action. We use MSE as loss and train for 100k steps with a batch size of 64. In test-time, given a task, similar to our proposed method, we use the same video model to generate subgoals for GCBC to complete the task.

**Diffusion Policy Goal-conditioned Behavior Cloning (DP GCBC).** For DP GCBC, we use an additional ResNet-18 image encoder to encode the goal image, and otherwise the model architecture is the same as DP BC. Note that the model for this baseline is also identical to the model we use in our proposed method. The goal image feature will then be concatenated with observation image features to condition the denoiser network. We set the diffusion denoising timestep to 100, horizon to 16, and train for 200k iterations with a batch size of 64.

**AVDC.** For AVDC, we directly adopt the official codebase (Ko et al., 2023). In Meta-world, we directly use the video model provided in the codebase, which is also the same video model for our method. In AVDC paper, the results are averaged across three different camera views, while our method only uses one camera view. Thus, for AVDC we report results on the same camera view as our method. In iTHOR, we notice that the video model resolution in the codebase is only (64, 64), which is too blurry to identify the target objects. Therefore, we train our video model with a resolution of (128, 128). Since AVDC does not experiment with Libero, we train our video model in this environment.

**SuSIE.** For SuSIE, we first train an image-editing diffusion model using the same video demonstration data to train the video model. The image-editing diffusion model also takes as input an observation image and a task description but outputs the next subgoal image. We then use this image-editing model to generate subgoals to guide our exploration, keeping all other hyperparameters the same.

