# OpenReview forum: "Grounding Video Models to Actions through Goal Conditioned Exploration"
_ICLR.cc/2025/Conference — ICLR 2025 Spotlight_

### Official Review · Reviewer_AfD8 · 2024-11-02

**Soundness:** 3
**Presentation:** 4
**Contribution:** 4
**Rating:** 8
**Confidence:** 3

**Summary:**

This paper proposes leveraging video guidance to learn goal-conditioned policies without access to any external supervision. By leveraging pre-trained task conditioning video generative models, they propose sampling a sequence of frames demonstrating how to complete a task. These frames serve as sub-goal supervision for training a downstream goal-conditioned policy. The policy is initialized via behavior cloning with random action bootstrapping, and then proceeds to improve by sampling the video model to generate more conditioning data for the policy. The self-supervised learning proceeds by collecting the rollouts in the replay-buffer and fine-tuning the policy on the replay-buffer. The video generative model is trained with just image pairs and no action labeled data and can be replaced with large internet-scale pre-trained models in the future.

Overall, this paper could be a significant contribution in combining self-supervised learning with video generative models to enable agents to solve complex tasks. But currently, there is confusion regarding the success of this method when random action bootstrapping fails. If addressed sufficiently, the rating can be increased.

**Strengths:**

The paper has several strengths including:
1. The work is well-motivated, well-written, and clear
2. The method is unique in proposing video models as a way to enable strong goal conditioning for the policy without needing action labels.
3. The idea of self-learning is unique compared to current RL approaches
4. Good set of ablations performed in the main paper and supplement to analyze the method well
5. The authors sufficiently addressed most of the limitations of their work. Despite these limitations, the method is still very interesting.

**Weaknesses:**

Some weaknesses of the paper include:
1. There is a lot of confusion around random action bootstrapping. Mainly, I am confused if it is possible to learn a policy without any successful actions (assuming random action bootstrapping does not result in any successful trajectories). Please see the first 6 points under ‘questions’ to know what needs to be added to the paper to address this weakness.
2. The cost of training this kind of model compared to BC is not mentioned in the limitation section. (The number of rollouts and iterations during self-training is high compared to the number of expert demonstrations for a BC policy).
3. A real world experiment is not provided. Seems that such an experiment would be high cost, maybe that is why it is not shown? Would be good to discuss if there is a way then to transfer easily from sim2real, but I acknolwedge sim2real is an existing open question, so possibly ok to not address this.
4. Unsure if 3.3 is a novel contribution. Diffusion policy generates action chunks already. What is the new ‘mean sampling’ proposing beyond the standard diffusion policy action generation scheme? Lines 239-242 are unclear.

**Questions:**

Key questions that should be addressed:
1. Maybe I have misunderstood, but how is it possible that without any ‘good’ actions ever given, the model can find actions that satisfy the goal eventually? Is it relying on the fact that at some point during random action bootstrapping there must be a few success cases? If so, how many successes in random action bootstrapping is required to learn the policy? Also, what is the performance of the policy with just random action bootstrapping (after line 3 of Algo 1) compared to performance after fine tuning with the video model in the self-supervised learning stage?
2. It is mentioned that random action bootstrapping is also performed at periodic intervals during training, but this is not indicated in Algorithm 1 (Algo 1 only shows random action bootstrapping before the loop starts). Please add that to the algorithm.
3. How is random action bootstrapping implemented in experiments? Specifically, what values are used a_low and a_high in equation 3 for all the different experiments? These details seem important to include in main or supplemental given that w/o random has a 0% success rate.
4. How frequently is the random action bootstrapping done? Would be a good point to add to the method (Sec 3.2) if this is a consistent number, or to the experiments if this is a hyperparameter.
5. Please provide details on how 'random extra' is performed. How many times extra do you do the random action bootstrapping?
6. For each experiment, it would be good to note the number of iterations of self-supervised exploration used (N in Algorithm 1).
7. If time permits, it would be good to understand the dependency of the policy during training on the model: What happens if the video model changes the sample rate it produces frames after the policy has been trained? For example, once the policy is trained, if the video model that is used changes to a pretrained model, does the policy produce viable actions? A simple experiment to run for this would be to just use the current trained policy, and replace the video model at test time, and see if the policy can still work.

Additional minor comments:
1. Would be good to include more details in the caption for Figure 2. What are these pairs of images showing (the start and goal image of the task or something else)?
2. Please mention the metric you use to evaluate the ‘success’ or ‘failure’ of a rollout? Is there some check in the simulation environment?
3. Typo line #147: should say ‘benefits both sides’
4. Typo line #423: should say ‘natural’ not ‘nature’
5. Line #314: Sure the video-model is zero-shot but the policy requires self-supervised training afterwards, so just change this sentence so it doesn't over-claim.

---

> ### Author Response · Authors · 2024-11-20
> **Reply to Reviewer AfD8 (1/5)**
>
> We thank reviewer AfD8 for their careful and constructive review. Please also see our response to each of the questions below.
> We have also updated the paper accordingly, where major updates include:
> 1. We conducted the suggested experiments on the policy robustness by using video models of different sampling rates at test time (see Appendix A.8);
> 2. We added thorough implementation details of random action bootstrapping along with hyperparameters for each environment in Appendix D.3.
> 3. We updated Algorithm 1 and the limitation section accordingly.
> 4. We fixed all the writing issues mentioned in the 'Additional minor comments' section.
>
>
>
> > W1. There is a lot of confusion around random action bootstrapping. Mainly, I am confused if it is possible to learn a policy without any successful actions (assuming random action bootstrapping does not result in any successful trajectories). Please see the first 6 points under ‘questions’ to know what needs to be added to the paper to address this weakness.
>
>
>
> Thank you for your valuable comments regarding the random action bootstrapping. Please see our detailed response to Q1-Q6 below.
>
>
>
>
>
> > W2. The cost of training this kind of model compared to BC is not mentioned in the limitation section.  (The number of rollouts and iterations during self-training is high compared to the number of expert demonstrations for a BC policy).
>
>
>
> We added the training cost difference in the limitation section of the updated manuscript.
>
>
>
>
>
>
>
> > W3. A real world experiment is not provided. Seems that such an experiment would be high cost, maybe that is why it is not shown? Would be good to discuss if there is a way then to transfer easily from sim2real, but I acknolwedge sim2real is an existing open question, so possibly ok to not address this.
>
>
>
> The focus of our work is to propose the core algorithm for grounding a video model, by which we can bridge the knowledge in foundational video model to executable actions. We have conducted extensive evaluation in multiple simulation environments and have shown the efficacy of the proposed method. As our approach makes no privileged knowledge about the simulation environment, this method can also directly be applied to real-world settings. However, due to noted computation costs above, we have not directly provide such an experiment.
>
>
> To deploy our sim policy to real-world scenarios, we believe that existing sim2real techniques can also be combined with our framework, such as Sys-ID and domain randomization.
>
>
>
>
>
>
>
>
> > W4. Unsure if 3.3 is a novel contribution. Diffusion policy generates action chunks already. What is the new ‘mean sampling’ proposing beyond the standard diffusion policy action generation scheme? Lines 239-242 are unclear.
>
>
>
> Previous works (e.g., Diffusion Policy) use action chunking in various common supervised learning settings, which is different from our problem setup. While action chunking has been used in some behavior cloning models, it is novel for the unsupervised exploration problem studied in this paper. We have also included a discussion on the contribution of the proposed action chunking for unsupervised exploration in our original manuscript (Lines 243-249).
>
>
> Lines 239-242 describe how our method samples an action chunk for the *random action exploration phase*, as noted in Line 238. Specifically, our method uses chunked actions in both random action exploration phase and the video-guided exploration phase. In the random action exploration phase, the action for exploration is directly sampled from a zero-mean uniform distribution; while in the video-guided exploration phase, the actions for exploration are from the diffusion policy's output.
>
> As described in Lines 240-242, 'action chunking' along with 'mean sampling' can ensure more consistent exploration (as repeatedly re-sampling and executing a single action from a zero-mean distribution can be highly unstable) and thus expand the coverage of the random exploration.

---

> > ### Author Response · Authors · 2024-11-20
> > **Reply to Reviewer AfD8 (2/5)**
> >
> > > Q1 (Part A). Maybe I have misunderstood, but how is it possible that without any ‘good’ actions ever given, the model can find actions that satisfy the goal eventually? Is it relying on the fact that at some point during random action bootstrapping there must be a few success cases? If so, how many successes in random action bootstrapping is required to learn the policy?
> >
> >
> >
> > The learning of our policy does not necessarily require or rely on successful action trajectories from random action bootstrapping. As shown in Table 5, the *w/o video* baseline (i.e., train the policy purely with random actions) fails, indicating that random action bootstrapping does not reach our desired goal states. Instead, the random action bootstrapping is only used to enable more effective video-guided exploration, and subsequent rounds of video-guided exploration will eventually be able to collect successful cases for training the policy.
> >
> > In our method, the intuition of random action bootstrapping is to collect data to learn the basic environment dynamics, because a random policy initialized from scratch cannot effectively process observed and goal images, hindering the video-guided exploration (as described in Lines 191-199).
> >
> >
> > As for why video-guided exploration can collect 'good' actions, the video model can provide helpful information to direct the exploration to some task-relevant state space and hence gradually improve the accuracy of the policy around these states by self-supervision. In inference time, the video frames then serve as the subgoals to finish the task. Since the dynamics to achieve the subgoals is learned during exploration, the policy then has the ability to successfully complete the task.
> >
> >
> >
> > Another analogue is that our method 'distills a video-diffusion model to a goal-conditioned policy for robot manipulation' as pointed out by Reviewer FoXm, with random action bootstrapping initializing the process.
> >
> >
> >
> >
> > > Q1 (Part B). Also, what is the performance of the policy with just random action bootstrapping (after line 3 of Algo 1) compared to performance after fine tuning with the video model in the self-supervised learning stage?
> >
> > We present the relevant ablation studies in Table 5 of our original submission. The policy with just random action bootstrapping (denoted by *w/o video*) completely fails (0% success rate), while combining random actions bootstrapping and video-guided exploration (denoted by *ours*) achieves 42.2% and 62.0% respectively.
> >
> >
> >
> >
> > > Q2. It is mentioned that random action bootstrapping is also performed at periodic intervals during training, but this is not indicated in Algorithm 1 (Algo 1 only shows random action bootstrapping before the loop starts). Please add that to the algorithm.
> >
> >
> > We have added the periodic random action bootstrapping part in Algorithm 1 of our updated manuscript.
> >
> >
> >
> >
> >
> >
> > > Q3. How is random action bootstrapping implemented in experiments? Specifically, what values are used a_low and a_high in equation 3 for all the different experiments? These details seem important to include in main or supplemental given that w/o random has a 0% success rate.
> >
> >
> >
> >
> > We sample random action chunks from uniform distribution. As described in Lines 238-242 of our original manuscript, we sample an action mean $a_m$ from a uniform distribution, and based on $a_m$, we sample a chunk of actions $a_c$ of length $l_c$ from Gaussian distribution, where the $i$-th action in the chunk is represented as $a_i^c \sim \mathcal{N}(a^m, \sigma)$.
> >
> > Specifically, in all tabletop manipulation environments, we set the length of the random action chunk $l_r$ to 24, and each random action episode consists of 5 chunks. Since the action in the navigation task is discrete, we do not leverage action chunking (a.k.a chunk size = 1) and each episode consists of 50 actions.
> >
> > We provide a thorough table of hyperparameters for random action bootstrapping in each environment in Appendix D.3 of the updated manuscript.
> >
> >
> > In the table below, we provide the values of $a_{\text{low}}$ and $a_{\text{high}}$ introduced in Section 3.2. The gripper action is discretized to -1 and 1 (close and open) and hence is not included in the table. In Libero environment, a smaller value in the orientation action space is set for more efficient random action bootstrapping.
> >
> > For the navigation environment, the action space is discrete: { Move Ahead, Turn Left, Turn Right, Done}, thus we directly do uniform sampling on these 4 actions.
> >
> >
> > |  | Libero | Meta-World | Calvin |
> > | --- | --- | --- | --- |
> > $a_{\text{low}}$ | [-1,-1,-1,-0.01,-0.01,-0.01] | [-1,-1,-1] | [-1,-1,-1,-1,-1,-1,-1] |
> > | $a_{\text{high}}$ | [1,1,1,0.01,0.01,0.01] | [1,1,1] | [1,1,1,1,1,1] |
> >
> > Please see Appendix D.3 of the updated manuscript for more details.

---

> > > ### Author Response · Authors · 2024-11-20
> > > **Reply to Reviewer AfD8 (3/5)**
> > >
> > > > Q4. How frequently is the random action bootstrapping done? Would be a good point to add to the method (Sec 3.2) if this is a consistent number, or to the experiments if this is a hyperparameter.
> > >
> > >
> > >
> > >
> > > The frequency $q_r$ to conduct periodic random exploration is set to 500 for all our experiments across all environments. (We haven't tuned this parameters and we believe that there might exist some more efficient values.)
> > >
> > > We provide a thorough table of hyperparameters for random action bootstrapping for each environment in Appendix D.3 of the updated manuscript and mentioned this in Section 3.2.
> > >
> > >
> > >
> > >
> > >
> > > > Q5. Please provide details on how 'random extra' is performed. How many times extra do you do the random action bootstrapping?
> > >
> > >
> > >
> > > In Table 5, 'random extra' denotes the experiment *with* periodic random action bootstrapping,
> > > while 'w/o extra rand' denotes the experiment *without* periodic random action bootstrapping (hence conducts random action bootstrapping only before the training starts, i.e., execute Line 3 and remove Lines 10-12 of the updated Algorithm 1).
> > >
> > > The implementation of how random actions are sampled is identical for both the random action bootstrapping before the training starts and the periodic random action bootstrapping during training, which is discussed in the answer of Q3 and Appendix D.3 of the updated manuscript.
> > >
> > > The frequency $q_r$ to do extra random exploration (a.k.a., the periodic random action bootstrapping) is set to 500 in the ablation study shown in Table 5 and also for all our experiments.
> > >
> > > Please see Table 15 in the updated manuscript for a full table of hyperparameters.
> > >
> > >
> > >
> > >
> > >
> > >
> > >
> > > > Q6. For each experiment, it would be good to note the number of iterations of self-supervised exploration used (N in Algorithm 1).
> > >
> > >
> > >
> > > The  number of iterations of self-supervised exploration training, $N$, is 200k for tabletop manipulation environments and 100k for visual navigation environments. These hyperparameters along with other training-time hyperparameters are also included in Appendix D.2 of our original submission.

---

> > > > ### Author Response · Authors · 2024-11-20
> > > > **Reply to Reviewer AfD8 (4/5)**
> > > >
> > > > > Q7. If time permits, it would be good to understand the dependency of the policy during training on the model: What happens if the video model changes the sample rate it produces frames after the policy has been trained? For example, once the policy is trained, if the video model that is used changes to a pretrained model, does the policy produce viable actions? A simple experiment to run for this would be to just use the current trained policy, and replace the video model at test time, and see if the policy can still work.
> > > >
> > > >
> > > >
> > > >
> > > > We conducted additional experiments, where we took a video model train at sampling rate of *7 frames to a goal* and tested its generalization to new sampling rates.
> > > >
> > > >
> > > >
> > > > Specifically, following the video model design in AVDC[1], we use the *Number of Frames to Goal* to denote the sampling rate of the video model.
> > > > For example,  *Number of Frames to Goal = 9* indicates that the video model is designed to uniformly generate 9 frames from the initial observation to the goal state (denoted by 9-frame video model for simplicity). That is, compared to a 7-frame video model, the sampling rate is denser (i.e., the temporal distance between two adjacent frames is smaller). We present the experiment results in the table below, where the policy is trained with a 7-frame video model (marked with an asterisk) while the number of frames to goal of the test-time video model varies from 5 to 9.
> > > >
> > > >
> > > >
> > > >
> > > >
> > > > | # Frames to Goal |  put-red-mug-left | put-red-mug-right | put-white-mug-left | put-Y/W-mug-right| Overall |
> > > > | -- | ----- | ----- | ----- |  ----- | ----- |
> > > > | 5 | 32.8 ± 12.0 |46.4 ± 10.3 |66.4 ± 5.4 |41.6 ± 4.8 |46.8 ± 8.1 |
> > > > | 6 | 38.4 ± 4.8 |48.0 ± 8.8 |68.8 ± 4.7 |36.0 ± 6.7 |47.8 ± 6.2 |
> > > > | 7* | 38.4 ± 15.3 |40.8 ± 7.8 |51.2 ± 3.9 |38.4 ± 8.6 |42.2 ± 8.9 |
> > > > | 8 | 39.2 ± 6.4 |51.2 ± 13.7 |65.6 ± 4.8 |25.6 ± 5.4 |45.4 ± 7.6 |
> > > > | 9 | 32.8 ± 5.3 |44.0 ± 8.8 |36.8 ± 9.9 |28.8 ± 5.9 |35.6 ± 7.5 |
> > > >
> > > >
> > > > | # Frames to Goal | put-choc-left | put-choc-right | put-red-mug-plate | put-white-mug-plate | Overall |
> > > > | -- | ----- | ----- | ----- |  ----- | ----- |
> > > > | 5  | 62.4 ± 4.1 | 76.8 ± 11.7 | 66.4 ± 4.1 | 59.2 ± 10.6 |66.2 ± 7.6 |
> > > > | 6  |  64.8 ± 6.9 | 72.0 ± 12.4 | 74.4 ± 9.3 | 58.4 ± 9.3 |67.4 ± 9.5 |
> > > > | 7*  | 70.4 ± 12.8 | 79.2 ± 3.9 | 72.8 ± 6.4 | 25.6 ± 11.5 |62.0 ± 8.7 |
> > > > | 8  |  52.0 ± 12.1 | 84.8 ± 8.2 | 68.0 ± 8.4 | 26.4 ± 10.9 |57.8 ± 9.9 |
> > > > | 9  |  59.2 ± 13.9 | 76.0 ± 8.8 | 74.4 ± 9.7 | 20.8 ± 7.3 |57.6 ± 9.9 |
> > > >
> > > >
> > > >
> > > >
> > > > The policy consistently achieves high performance across video models with different sampling rates, demonstrating its robustness to variations in the sampling rate during testing.
> > > > This is probably because of the robust policy training enabled by the combination of random action bootstrapping and video-guided exploration.
> > > >
> > > > Please refer to A.8 of the updated manuscript for more discussion.
> > > >
> > > >
> > > >
> > > > [1] Learning to Act from Actionless Videos through Dense Correspondences

---

> > > > > ### Author Response · Authors · 2024-11-20
> > > > > **Reply to Reviewer AfD8 (5/5)**
> > > > >
> > > > > Response to the additional minor comments:
> > > > >
> > > > > > 1. Would be good to include more details in the caption for Figure 2. What are these pairs of images showing (the start and goal image of the task or something else)?
> > > > >
> > > > >
> > > > >
> > > > > We added more description in the caption of Figure 2 in the updated manuscript.
> > > > >
> > > > > For (a), the images show the goal object state of a subset of our Libero tasks. For example, the top-left image corresponds to the task 'put the red mug to the left plate'.
> > > > >
> > > > > For (b), the images show the start observation of the tasks in Meta-World.
> > > > > For instance, the top-left image of (b) corresponds to the task 'Door Open'.
> > > > >
> > > > > For (c), the images show the goal object state in Calvin.
> > > > > For example, the top-left image of (c) corresponds to the task 'move slider left'.
> > > > >
> > > > > For (d), the images are the rendered agent view at some specific locations in the scenes. Images here are to give readers a broad view of the scene layouts.
> > > > >
> > > > >
> > > > >
> > > > >
> > > > > > 2. Please mention the metric you use to evaluate the ‘success’ or ‘failure’ of a rollout? Is there some check in the simulation environment?
> > > > >
> > > > >
> > > > >
> > > > > Yes. The simulation environment will check the state of the target object in manipulation tasks or check the agent state in visual navigation tasks after executing an action.
> > > > >
> > > > > For manipulation tasks, if the target object is within the range of the specified goal state, the environment will return a success; for navigation tasks, if the target object is within a certain distance and in the view of the agent and the agent executes a 'Done' action, the environment will return a success.
> > > > >
> > > > > During a rollout, if a success is returned, this rollout will be counted as success and terminated; otherwise, if we have finished a rollout (i.e., have sequentially executed all the synthesized video frames) and no success is returned, this episode will be counted as a failed rollout.
> > > > >
> > > > > We have added the corresponding description in Appendix D.4 of our updated manuscript.
> > > > >
> > > > >
> > > > >
> > > > >
> > > > > > 3-5. Typos and Writing
> > > > >
> > > > > We have updated the paper accordingly along with some other minor typos, highlighted by the blue text.

---

> > > > > > ### Comment · Reviewer_AfD8 · 2024-12-03
> > > > > >
> > > > > > Overall, the authors did a very good job of clarifying my doubts. Therefore, I update my rating to an 8 (accept, good paper).

---

> > > > > > > ### Author Response · Authors · 2024-12-03
> > > > > > > **Official Comment by Authors**
> > > > > > >
> > > > > > > Dear Reviewer AfD8,
> > > > > > >
> > > > > > > We are truly grateful for your detailed and insightful feedback and deeply appreciate your positive assessment of our manuscript. Thank you again for your time and effort in reviewing our manuscript.
> > > > > > >
> > > > > > > Best,
> > > > > > > Paper Authors

---

> ### Author Response · Authors · 2024-12-02
> **Official Comment by Authors**
>
> Dear Reviewer AfD8,
>
> We deeply appreciate the time and effort you have dedicated to reviewing our work. We understand that this may be a particularly busy period, but we would like to kindly send a gentle reminder that the end of the reviewing process is approaching. We have provided detailed responses to your previous questions and suggestions -- please let us know if you have any additional concerns.
>
> Thank you again for reviewing our work and we look forward to hearing from you soon.
>
> Thanks,
> Paper Authors

---

### Official Review · Reviewer_FoXm · 2024-11-03

**Soundness:** 3
**Presentation:** 3
**Contribution:** 3
**Rating:** 6
**Confidence:** 3

**Summary:**

This paper introduces a method to "distill" a video-diffusion model to a goal-conditioned  policy for robot manipulation. The authors proposed to leverage the conditional generation capability of video generative models to sample future trajectories as goals for refining the goal-conditioned policy, which can gradually align its action prediction with the actual ground truth. The resulting method achieves state-of-the-art results on several commonly-used benchmarks.

**Strengths:**

The paper is clearly written and easy to follow, the method proposed achieves state-of-the-art results on several commonly used datasets with consistent improvement. The ablation study does show the effect of each designed module.

**Weaknesses:**

One concern about the proposed method lies in the selection of video-guidance. As assumed in this paper, a good enough conditional video generative model is key in improving the goal-conditioned policy in the iterative refining process. This leads to questions on the availability of such models for specific tasks with limited demonstrations. Though the authors mentioned leveraging pre-trained text-to-video models could be discussed in the future, it seems necessary even at the current scope (or if there is other work arounds or model performance guarantees on the data needed for a good enough task specific video generator).

**Questions:**

See the weakness section.

---

> ### Author Response · Authors · 2024-11-20
> **Reply to Reviewer FoXm**
>
> We thank Reviewer FoXm for the careful and constructive review. Please see our response to the listed concerns below.
>
> > Weakness: One concern about the proposed method lies in the selection of video-guidance. As assumed in this paper, a good enough conditional video generative model is key in improving the goal-conditioned policy in the iterative refining process. This leads to questions on the availability of such models for specific tasks with limited demonstrations. Though the authors mentioned leveraging pre-trained text-to-video models could be discussed in the future, it seems necessary even at the current scope (or if there is other work arounds or model performance guarantees on the data needed for a good enough task specific video generator).
>
>
>
> Thank you for your valuable comments. As our policy follows the subgoals given by the video frames, we agree that a good video model is important for completing the given tasks.
>
>
> We would like to clarify that we can actually obtain a good conditional video generation model from scratch in the domain of a very limited amount of data in the absence of pretrained video models. In Appendix A.1, our conditional video generation trained with only 20 demonstrations outperforms BC trained with 50 demonstrations. In Appendix A.2, we further decrease the training data for the conditional video generation model to 10 demonstrations per task, and our method still can achieve non-trivial performance. In all our experiments in the main paper, we also illustrate how in the *absence of an existing pre-trained video model*, where we train a conditional video generation model from scratch with only 20 demonstrations, already significantly outperforms baselines.
>
>
> With the prior dynamic knowledge learned in the pre-trained video model, we believe that adapting or fine-tuning from an existing pre-trained text-to-video model would only serve to further improve the performance of our approach and reduce the reliance on the downstream in-domain robotics data.
>
>
> In addition to leveraging pre-trained video models, there are also several additional ways that could be used to improve the video model's generalizability to scenarios with limited demonstrations that are orthogonal to our work, but which we would be happy to discuss in future work such as:
>
> 1. Some existing robotic data generation engine, such as MimicGen[1], can be leveraged to generate more demonstration data for training or even generate new tasks for the video model (GenSim[2]).
> 2. Foundation models can be incorporated to enhance the model's generalizability to unseen scenarios. For example, a VLM can be used to select promising videos out of a batch of video candidates generated by the video model in settings that are out-of-distribution.
> 3. Using the environment interaction feedback from the workspace to correct and fine-tune the current video model is another potential direction to improve the model performance.
>
> While we believe these are interesting directions for future work, we believe that our paper is already interesting to the broader community as it provides a new way to convert video models to actions to execute, without needing any difficult to obtain in domain action labels. With the large recent focus on scaling video models and existing power of large foundation video models, we believe pretrained video models will be useful for many robotics tasks, with a key bottleneck being how we can actually convert videos to continuous actions to execute, which we tackle in this paper.
>
> [1] MimicGen: A Data Generation System for Scalable Robot Learning using Human Demonstrations
>
> [2] GenSim: Generating Robotic Simulation Tasks via Large Language Models

---

> > ### Comment · Reviewer_FoXm · 2024-12-03
> > **Post-rebuttal comment**
> >
> > Thanks the authors for the clarification and looking forward to the future extension of this work, I will keep my original positive rating.

---

> > > ### Author Response · Authors · 2024-12-03
> > > **Official Comment by Authors**
> > >
> > > Dear Reviewer FoXm,
> > >
> > > We are truly grateful for your detailed and insightful feedback and deeply appreciate your positive assessment of our manuscript. Thank you again for your time and effort in reviewing our manuscript.
> > >
> > > Best,
> > > Paper Authors

---

### Official Review · Reviewer_beJr · 2024-11-03

**Soundness:** 3
**Presentation:** 4
**Contribution:** 4
**Rating:** 8
**Confidence:** 3

**Summary:**

This paper presents a framework for grounding large pretrained video models to actions within an embodied environment through self-exploration. By generating visual goals from video states, the authors propose a goal-conditioned exploration strategy that allows agents to solve complex tasks without needing external supervision like rewards or action labels. A set of methods, including chunked action prediction and exploration with randomized exploration, are proposed to enable robust exploration. The proposed method is validated on several simulated environments, such as Libero, MetaWorld, Calvin, and iTHOR, where it demonstrates performance on par with or exceeding that of behavior cloning baselines trained on expert demonstrations.

**Strengths:**

1. The unsupervised grounding of video models to actions eliminates the dependency on expensive action annotations, efficiently addressing the problem of mapping video-based observations to actionable policies.
2. The proposed method achieves strong performances across multiple evaluation environments, outperforming supervised methods in quantitative and qualitative resutls. Besides, the method show adaptability across different domains, from robotic manipulation to visual navigations, demonstrating the robustness and generalization ability.
3. The ablation studies are comprehensive, demonstrating the effectiveness of the each proposed components.
4. Video-guided exploration allows the agent to focus on task-relevant state spaces, resulting in more targeted exploration and more effective data collection. This targeted approach enables the model to gather high-quality training data efficiently, especially for complex, long-horizon tasks that would otherwise require extensive manual annotation.

**Weaknesses:**

This paper is well-written with strong motivation and comprehensive evluation results. I only have some minor weakness and questions.

1. The reliance on random exploration may not be able to achieve high performance in tasks requiring high precision, such as tasks involving fine-grained manipulation or exact positioning. This approach may struggle to find optimal actions in environments where precise control is crucial, limitations is applications.
2. The proposed method is highly rely on the quality and generalization ability of the pretrained video generation models. If these models fail to generalize well to new environments, the effectiveness of the approach could be significantly constrained, especially in dynamic or unfamiliar settings.
3. The computational requirements might be high due to the chunk-level action prediction.
4. In algorithm 1, line 10-12 lacks an indentation, making it hard to read what is the condition and what is the content of the end-if sentence.

**Questions:**

1. Would the computational efficiency be improved by using distilled video diffusion models, which could generate videos in a very fast manner? Would the distilled models affect the performance of the proposed method?
2. Given the reliance on the pretrained video generation model, how does the framework handle cases where the video model’s generalization is limited? Are there specific techniques that could be employed to improve the model’s adaptability to novel or dynamic environments?

---

> ### Author Response · Authors · 2024-11-20
> **Reply to Reviewer beJr (1/2)**
>
> We thank Reviewer beJr for the positive and careful review. Please see our response to each listed concern below.
>
> > W1. The reliance on random exploration may not be able to achieve high performance in tasks requiring high precision, such as tasks involving fine-grained manipulation or exact positioning. This approach may struggle to find optimal actions in environments where precise control is crucial, limitations is applications.
>
>
> Thank you for your valuable comments. We agree that achieving high-precision configurations can be challenging through pure random action sampling.
> In such scenarios, it would be important to further have exploration primitives that can help us achieve these high-precision goals (which we discuss in the limitation part of Section 5 and Appendix C.2 of the paper).
>
>
>
>
>
>
> > W2. The proposed method is highly rely on the quality and generalization ability of the pretrained video generation models. If these models fail to generalize well to new environments, the effectiveness of the approach could be significantly constrained, especially in dynamic or unfamiliar settings.
>
>
> We agree that our approach relies on the quality and generalization ability of the pretrained video generation. In dynamic or unfamiliar settings, we believe there are several future ways to improve the video model's performance which we believe is orthogonal to our current work.
>
> 1. Foundation models can be incorporated to enhance the model's generalizability to unseen scenarios. For example, a VLM can be used to select an accurate video from  a batch of video candidates generated by the video model in settings that are out-of-distribution.
> 2. Adapting or fine-tuning our video model from existing pre-trained video generative models with a small amount of in-domain data can also enhance the adaptability to novel or dynamic environments.
> 3. Using the collected environment interaction feedback from the workspace to correct and fine-tune the video model is another potential direction to improve the model performance.
>
>
>
>
>
>
>
>
> > W3. The computational requirements might be high due to the chunk-level action prediction.
>
> Compared to single-action model, predicting a chunk of actions increases the output dimension of the policy model, however, it reduces the number of model forward passes (since the model can generate a sequence of actions to execute with only one model forward).
>
> In practice, we use diffusion policy as our goal-conditioned policy, which predicts a chunk of actions. Following the diffusion policy paper, we set the number of diffusion timestep 100, but we use DDIM to accelerate sampling during inference, requiring only 8 denoising steps. Therefore, the computation requirements of chunk-level action prediction are similar to a single action prediction model.
>
>
>
>
>
>
> > W4. In algorithm 1, line 10-12 lacks an indentation, making it hard to read what is the condition and what is the content of the end-if sentence.
>
>
> Sorry for the confusion caused.
>
> In Algorithm 1, Lines 5-9 are the if-statement for video-guided exploration, which only executes with frequency $q_v$. We use the 'end if' in Line 9 to mark the end of the video-guided exploration block.
>
> The policy training code (Lines 10-12 of the original manuscript, Lines 13-16 of the updated manuscript) are supposed to execute in every iteration of the For-Loop, therefore, we put them outside of the if-statement (below 'end if').
>
> In the updated manuscript, we appended additional comments to further enhance the readability of Algorithm 1.

---

> > ### Author Response · Authors · 2024-11-20
> > **Reply to Reviewer beJr (2/2)**
> >
> > > Q1. Would the computational efficiency be improved by using distilled video diffusion models, which could generate videos in a very fast manner? Would the distilled models affect the performance of the proposed method?
> >
> >
> >
> > A distilled video diffusion model will likely enhance the computational efficiency, since distillation of diffusion model can usually reduce the model size and the test-time denoising steps.
> >
> > We believe that using such models would not significantly affect the performance of our model, because the high-level guidance knowledge that synthesizes frames to complete the given tasks will also be transferred to the distilled video model.
> >
> >
> > More generally, our proposed method can be combined with any generative model that is able to generate feasible frames towards the completion of the given tasks, including video models or image inpainting models (as shown in *ours w/ SuSIE* in Table 1).
> >
> >
> >
> >
> > > Q2. Given the reliance on the pretrained video generation model, how does the framework handle cases where the video model’s generalization is limited? Are there specific techniques that could be employed to improve the model’s adaptability to novel or dynamic environments?
> >
> >
> > In unfamiliar or dynamic settings, we believe there are several future ways to improve the video model's performance:
> >
> > 1. Foundation models can be incorporated to enhance the model's generalizability to unseen scenarios. For example, a VLM can be used to select an accurate video from a batch of video candidates generated by the video model in settings that are out-of-distribution.
> > 2. Adapting or fine-tuning our video model from existing pre-trained video generative models with a small amount of in-domain can also enhance the adaptability to novel or dynamic environments.
> > 3. Using the collected environment interaction feedback from the workspace to correct and fine-tune the video model is another potential direction to improve the model performance.

---

### Author Response · Authors · 2024-11-28
**General Response by Authors**

Dear Reviewers,

We sincerely thank all reviewers for the detailed reviews as well as positive and constructive feedback. As the discussion period is drawing to a close, we would like to summarize the changes we have made to our paper:

1. We conducted experiments on policy robustness by using video models of different sampling rates at test time. Experiment results and analysis are detailed in Appendix A.8.
2. We provided comprehensive implementation details of random action bootstrapping with the specific hyperparameters used for each environment. These details can be found in Appendix D.3.
3. We added the description of the task success metric in Appendix D.4.
4. We revised Algorithm 1 accordingly to improve clarity.
5. We fixed all the identified typos and updated the manuscript in line with each writing suggestion.

We believe that these revisions along with the per reviewer response have adequately addressed each listed concern. We deeply appreciate your time and effort in reviewing our manuscript and reading our response. Your feedback is invaluable for us to improve our manuscript. Please let us know if you have any further questions -- we would be very happy to  discuss and address them.



Thanks,
Paper Authors

---

### Meta-Review · Area_Chair_emBH · 2024-12-19

**Metareview:**

The paper proposes a novel approach to grounding video models to actions through goal-conditioned exploration, which is a good contribution to the field of embodied AI. The authors have addressed most of the concerns raised by the reviewers in a satisfactory manner.  The proposed method has shown good performance and potential for further development. Minor remaining issues do not outweigh the paper's strengths and contributions. Therefore, the paper is recommended for acceptance.

**Additional Comments On Reviewer Discussion:**

Reviewer 1 (beJr): Raised concerns about the reliance on random exploration for high-precision tasks, the dependence on the quality and generalization ability of the pretrained video generation models, the high computational requirements, and a formatting issue in the algorithm. The authors acknowledged the limitations of random exploration in high-precision tasks and discussed future exploration primitives. They also proposed several ways to improve the video model's performance in dynamic or unfamiliar settings. Regarding the computational requirements, they compared chunk-level action prediction to single-action models and explained the use of diffusion policy to reduce computation. The algorithm formatting issue was addressed by adding comments for better readability.

Reviewer 2 (FoXm): Questioned the availability of good conditional video generative models for specific tasks with limited demonstrations and the novelty of some contributions. The authors demonstrated that a good conditional video generation model can be obtained with a very limited amount of data. They also clarified the novelty of their approach compared to previous works and discussed future directions to improve the video model's generalizability.
Reviewer 3 (AfD8): Identified confusion around random action bootstrapping, the lack of mention of training cost compared to BC, the absence of a real-world experiment, and an unclear contribution in one aspect. The authors conducted experiments on policy robustness, added detailed implementation details of random action bootstrapping, updated the algorithm and limitation section, and fixed writing issues. They also discussed the role of random action bootstrapping and the learning process of the policy. Regarding the training cost, they added it to the limitation section. They explained the focus on simulation environments and the potential for applying existing sim2real techniques. They clarified the novelty of the proposed action chunking and its role in the exploration process.

---

### Decision · Program_Chairs · 2025-01-22

Accept (Spotlight)